# Brain mitochondrial diversity and network organization predict anxiety-like behavior in male mice

Ayelet M. Rosenberg [1], Manish Saggar [2], Anna S. Monzel[1], Jack Devine[1], Peter Rogu[3], Aaron Limoges[4,5], Alex Junker[1], Carmen Sandi [6], Eugene V. Mosharov[7,8], Dani Dumitriu [3,9,10], Christoph Anacker [3,5,8] & Martin Picard [1,8,11,12] ✉

The brain and behavior are under energetic constraints, limited by mitochondrial energy transformation capacity. However, the mitochondria-behavior relationship has not been systematically studied at a brain-wide scale. Here we examined the association between multiple features of mitochondrial respiratory chain capacity and stress-related behaviors in male mice with diverse behavioral phenotypes. Miniaturized assays of mitochondrial respiratory chain enzyme activities and mitochondrial DNA (mtDNA) content were deployed on 571 samples across 17 brain areas, defining specific patterns of mito-behavior associations. By applying multi-slice network analysis to our brain-wide mitochondrial dataset, we identified three large-scale networks of brain areas with shared mitochondrial signatures. A major network composed of cortico-striatal areas exhibited the strongest mitochondria-behavior correlations, accounting for up to 50% of animal-to-animal behavioral differences, suggesting that this mito-based network is functionally significant. The mito-based brain networks also overlapped with regional gene expression and structural connectivity, and exhibited distinct molecular mitochondrial phenotype signatures. This work provides convergent multimodal evidence anchored in enzyme activities, gene expression, and animal behavior that distinct, behaviorally-relevant mitochondrial phenotypes exist across the male mouse brain.

The shaping of behaviors by life experiences is driven by energetically demanding circuitry across the brain[1]. The brain's enormous energetic demand is mainly met by ATP produced through oxidative phosphorylation (OxPhos), subserved by the combined activities of respiratory chain (RC) enzymes within mitochondria[2]. As a result, mitochondria influence multiple aspects of brain development and cell biology ranging from dendritic and axonal branching[3,4], remodeling of gene expression[5], neurotransmitter release and excitability in mature synapses[6–8], neurogenesis[9], and inflammation[10]. The mounting molecular and functional evidence that the brain is under energetic constraints[11,12] suggests that if we want to understand the basis of brain

function and behavior, we must understand key aspects of brain mitochondrial biology.

Mitochondria are small, dynamic, multifunctional organelles with their own genome[13], but not all mitochondria are created equal. Mitochondria serving different cellular demands (i.e., in different cell types) have different relative molecular compositions, morphologies, and functional phenotypes[14–19]. Therefore, developing a comprehensive understanding of the association between mitochondrial biology and animal behavior calls for assessments of multiple functional and molecular mitochondrial features, across multiple brain areas. By analyzing mitochondrial features across multiple brain areas simultaneously, we

can also potentially uncover unknown brain "mitochondrial networks". This idea aligns with the evolving understanding of large-scale brain circuitry and metabolism[11,20], and of the network distribution of neural activity achieved using brain-wide, high-spatial resolution methods (e.g., MRI-based functional and structural connectivity maps)[21,22]. Although a similar degree of resolution for mitochondrial phenotyping is not feasible with current technologies, the miniaturization of biochemical and molecular mitochondrial assays open new possibilities to systematically map mitochondria-to-behavior associations across multiple cortical and sub-cortical brain areas in the same animal.

Mitochondrial biology and animal behaviors measured on standardized tests exhibit both naturally occurring and acquired (i.e., experience-dependent) variations, providing an opportunity to map their associations. For example, behaviorally, exposure to social stress such as chronic social defeat increases potential threat vigilance and social avoidance[23], but animals differ in their vulnerability/resilience to these stress-induced behavior changes[24,25]. Experimental challenges aimed at modeling neuroendocrine disturbances resulting from chronic stress, such as chronic exposure to corticosterone, also induce avoidance behaviors associated with recalibrations in specific brain circuitry[26], gene expression[27,28], and anatomical plasticity (i.e., atrophy) in stress-sensitive brain areas like the hippocampus[29]. In relation to mitochondria, a separate body of literature similarly documents naturally occurring mitochondrial variation[30-32], as well as stress-induced functional mitochondrial recalibrations that occur within days to weeks of stress exposure[33-35] (meta-analysis in ref. [36]), potentially linking mitochondrial biology to behavior. More direct causal experiments show that mitochondrial RC enzyme activities directly influence the brain and specific behavioral domains, including working memory[37,38], social dominance[31,39,40] and anxiety-related behavior[4]. Targeted mitochondrial defects even cause mood disorder-like phenotypes in animals[41], positioning mitochondria as upstream modulators of brain function and behavior. Moreover, mitochondrial RC defects are likely implicated in the etiology of psychiatric, neurological and degenerative disorders in humans[42], and in vivo brain metabolic imaging studies show that energy metabolism in specific brain areas (e.g., nucleus accumbens, NAc) predict cognitive performance and anxiety[43-45], making these biological questions also potentially relevant to mental health in humans.

Although the importance of mitochondria for brain structure and function is unequivocal, we lack an understanding of potential biochemical and other functional differences in mitochondria across different brain areas. This calls for more systematic mapping of area-specific brain mitochondrial biology in relation to behavior. To address the hypothesis that mitochondrial phenotypes in specific brain areas are associated with behaviors, we have further miniaturized existing biochemical and molecular assays of mitochondrial OxPhos enzyme activities for sub-milligram tissue samples and deployed them across 17 cortical and sub-cortical brain areas in mice with a wide range of behavioral phenotypes, quantified through four behavioral tests. Using network-based connectivity analysis, plus brain-wide gene expression data from the Allen Mouse Brain Atlas, we also sought to explore the distribution of mitochondrial phenotypes across brain areas, finding evidence that mouse brain mitochondria specialize as distinct functional networks linked to behavior. Further, we use gene co-expression and structural/anatomical connectivity data, which anchor the newly observed mitochondrial circuits into other modalities, and provide a foundation for future mechanistic studies to elucidate the specific link between mitochondrial biology, distributed brain networks, and behavioral variation.

## Results
### Miniaturization of mitochondrial assays for sub-milligram resolution
Although enzymatic activity assays directly reflect the capacity of RC complexes and therefore energy production capacity, previously available assays suffer from low throughput that preclude a multi-area, brain-wide analysis in dozens of animals. Building from our efforts to miniaturize and scale the throughput of mitochondrial RC activity assays in immune cells[30,32], here we miniaturized and optimized spectrophotometric assays in 96-well plate format for brain tissue, validated against the standard cuvette-based reaction (Supplementary Fig. 1, and *Methods*). Using this optimized platform, we can quantify the enzymatic activities of RC complex I (CI, NADH-ubiquinone oxidoreductase), complex II (CII, succinate-ubiquinone oxidoreductase), complex IV (CIV, cytochrome c oxidase), and citrate synthase (CS, a Krebs cycle enzyme and marker of mitochondrial content) in two 1mm-diameter, 200μm deep tissue punches. This represents <1 mg of tissue (estimated 0.33 mg), over an order of magnitude more sensitive than currently available methods, and represents, in our hands, the lowest detection limit for mouse brain tissue.

In this sample processing pipeline, the same biological sample used for enzymatic activities is also used to quantify mtDNA abundance, both by i) the classical metric relative to the nuclear genome – mtDNA/nDNA ratio, termed mtDNA copy number (*mtDNAcn*)[46], and ii) per tissue volume (mtDNA copies per μm$^3$ or per mg), termed *mtDNA density*. qPCR data of nuclear genome abundance across the brain, independent of the mtDNA, showed that cellular density varies by up to ~8.5-fold between brain areas (highest: cerebellum, lowest: visual cortex). Unlike in peripheral tissues, this remarkably large variation in neuronal and glial cell somata density significantly skews mtDNAcn estimates. Supplementary Fig. 2 shows the inter-correlations of mtDNAcn and mtDNA density in relation to enzyme activities. Compared to mtDNAcn, which is confounded by the presence or absence of somata/nuclear genome, *mtDNA density* was more consistently associated with RC enzymatic mitochondrial phenotypes across all brain areas, and therefore likely represents a more generalizable estimate of mitochondrial genome abundance across mouse brain areas.

We subsequently combined these five primary mitochondrial measures into a mitochondrial health index (MHI) by dividing RC activities (CI + CII + CIV) by mitochondrial content (CS+mtDNA density), thus creating an index of energy transformation capacity on a per-mitochondrion basis[30] (see *Methods* for details). Although each of the six resulting features are partially correlated with other features, the proportion of shared variance between individual features across brain areas is 31–64%, indicating that they each contribute some non-redundant information about mitochondrial phenotypes, which can be deployed at scale.

Our study design first aimed to profile animal-to-animal differences in mitochondrial phenotypes across a broad set of brain areas known to be associated with anxiety-related behavior, social behaviors, cognition, or mitochondrial disorders, and to relate these measures to each animal's behavioral phenotypes. In total, we enzymatically and molecularly phenotyped 571 samples covering 17 cortical, sub-cortical, and brainstem brain areas isolated by bilateral punches at defined stereotaxic coordinates (Table 1 and Supplementary Fig. 3). To eventually compare the specificity of our findings related to brain and behavior, 5 peripheral tissues were also collected and analyzed from each animal, which we expected to show somewhat related but more modest associations with behavioral outcomes.

### Protein levels and enzymatic activity
We initially explored if RC protein abundance is a viable surrogate for mitochondrial RC activity[31], which could theoretically allow high-spatial resolution imaging of the entire brain. We focused on the cerebellum due to its well-defined cellular composition and laminar organization, where the Purkinje cell layer is flanked by molecular and granular layers[14] (Supplementary Fig. 4a). Biologically, compared to protein abundance, *enzymatic activity* ultimately determines mitochondrial RC function and energy transformation capacity and consequently should be regarded as the most representative measure of

**Table 1 | Expanded abbreviations and Bregma coordinates for each brain area (n = 17, top) and non-brain tissues (n = 5, bottom)**

| Abbreviation | Area name | Bregma/Anatomical location |
|---|---|---|
| mOFC | Medial orbitofrontal cortex | 2.42 |
| mPFC | Medial prefrontal cortex | 1.22 |
| CPu | Caudoputamen | 1.22 |
| NAc | Nucleus accumbens | 1.22 |
| M1 | Primary motor cortex | 1.22 |
| Hypoth | Hypothalamus | −0.48* |
| Thal | Thalamus | −0.68* |
| DGd | Dorsal dentate gyrus | −1.78 |
| Amyg | Amygdala | −1.78 |
| CA3 | CA3 region | −2.58 |
| VTA | Ventral tegmental area | −2.58 |
| V1 | Primary visual cortex | −2.58 |
| SN | Substantia nigra | −3.08 |
| DGv | Ventral dentate gyrus | −3.38 |
| PAG | Periaqueductal gray | −3.78* |
| Cereb | Cerebellum | −6.38 |
| VN | Vestibular nucleus | −6.38 |
| AG | Adrenal glands | Both glands pooled |
| Liver | Liver | Inferior portion of left lobe |
| Heart | Heart | Inner wall of left ventricle |
| Soleus | Red oxidative skeletal muscle | Tendon-to-tendon, right leg |
| WG | White glycolytic skeletal muscle | Superficial medial head, right leg |

All areas were taken bilaterally except those marked with *, which were obtained by collecting tissue from two consecutive slices.

mitochondrial phenotypes. In consecutive cerebellar slices, we compared RC complex II *enzymatic activity* measured spectrophotometrically, to the *protein abundance* of a complex II subunit, SDHA (succinate dehydrogenase, subunit A), for which a validated high-affinity antibody allows its quantification by microscopy (Supplementary Fig. 4b, c). Across the three cerebellar layers, enzyme activity did not correlate with protein abundance assessed by immunohistochemistry and densitometry (proportion of shared variance, $r^2 = 0.02{-}0.07$), indicating that protein abundance and enzymatic activity are not equivalent (Supplementary Fig. 4d). The reasons for this finding could include the action of post-translational modifications, variation in the stoichiometry of the four SDH subunits, or the biochemical context that drive biochemical activity independent of protein content (e.g., ref. [47]). Therefore, we focus all downstream analyses on direct biochemical measures of mitochondrial RC enzymatic activity and mtDNA density.

### Diversity of mitochondrial RC activities between animals

We next examined mitochondria-behavior associations in a cohort of mice exhibiting naturally occurring behavioral and mitochondrial variation. Our goal was to identify robust and generalizable associations between mitochondrial phenotypes and behavior, rather than potential correlations that would exist among only subgroups of animals (naïve or stressed). Therefore, to further extend the spectrum of mito-behavior variation using well-characterized rodent stress models, subgroups of the cohort were either chronically administered corticosterone (CORT)[48] for three months or exposed to 10 days of chronic social defeat stress (CSDS)[49] (see Supplementary Fig. 3a). To create additional diversity among groups, half of the CSDS mice were allowed to recover

for two months, as some stress-induced behavioral and mitochondrial changes may change again when the stressor is removed. Based on previous work[24,50], we also expected naturally occurring behavioral and brain molecular phenotype differences between animals that are resilient or susceptible to CSDS (based on the social interaction test; see *Methods*)[51]. Together, the naturally occurring variation in mitochondria and behavior plus the effects of various exposures provides a strong test of robustness for our hypothesis, which was that across a diverse population of mice, mitochondrial phenotypes in specific brain areas are consistently associated with behaviors.

Across our cohort of inbred male mice with a range of exposures, a wide spectrum of mitochondrial phenotypes was observed. The average variation for all measures (4 individual enzymatic activities and mtDNA density) across all animals and all 17 brain areas was a C.V. of 36% (coefficient of variation = standard deviation/mean). For any given brain area, the absolute variation in mitochondrial phenotypes between mice reached up to 2.9-fold between the animal with the lowest and the animal with the highest activities. This means that for a given brain area, there are large mouse-to-mouse differences in mitochondrial content and RC activities, even among inbred naïve mice not exposed to stressors (Supplementary Fig. 5). Peripheral tissues showed an average animal-to-animal C.V. of 25%, about a third less variation than for brain areas, indicating that the brain may exhibit particularly large inter-individual differences.

To verify that the interventions used to enhance naturally occurring variation in behavior and mitochondria were effective, we first compared mitochondrial features in mice exposed to CORT and CSDS relative to naïve mice. The effects of CORT and CSDS on mitochondrial phenotypes were quantified as standardized effect sizes (Hedges g) and shown in Fig. 1a–c (detailed in Supplementary Fig. 6). Both interventions altered mitochondrial RC complexes, CS enzymatic activities, mtDNA density, and MHI in an area-specific manner. Although not statistically significant, CORT-treated mice tended to have *higher* mitochondrial activities than non-stressed animals in ~60% of brain areas, whereas CSDS animals trended towards *lower* activities in ~82% of brain areas, which was statistically significant for CI, CIV, and MHI measures (Fig. 1d), suggesting opposing effects of these two different stress models on brain mitochondria. This could be due to either the nature of the stressor, or their durations. The study was not powered to detect significant differences across all mitochondrial parameters, brain areas, and experimental conditions, so we report here the effect sizes and unadjusted *p*-values. The amygdala (Amyg) showed the greatest CORT-induced increase in CII enzymatic activity (+49%, $p = 0.03$, unadjusted *p*-value), whereas the periaqueductal gray (PAG) showed the largest CSDS-induced decrease in CI activity (−42%, $p = 0.02$, unadjusted, see Fig. 1a).

To determine if brain areas were co-regulated in their stress-induced mitochondrial recalibrations, we employed a topological data analysis (TDA)-based Mapper approach. *Mapper* is a variant of non-linear dimensionality reduction methods (manifold learning) that produces a graph or network embedding of the high-dimensional data (a.k.a. shape graph) while recovering projection loss using an additional partial clustering step[52,53]. When applied to the six mitochondrial features (i.e., four enzyme activities, mtDNA density, and MHI) separately for CORT and CSDS (both measured as delta of stress vs. the naïve group average), the shape graphs revealed differences in regional mitochondrial recalibrations across the two groups (Fig. 1e). CORT-induced mitochondrial recalibrations were relatively more area-specific or segregated, whereas CSDS caused a more coherent or integrated mitochondrial response across all brain areas (Fig. 1f). This was quantified using a graph-theoretical measure of participation coefficient (PC)[54], where higher values of PC indicate uniformly distributed connectivity across areas (integrated network) and lower values indicate a segregated connectivity pattern. The CORT group exhibited a ~25% lower PC than the CSDS group ($p < 0.05$, Fig. 1g).

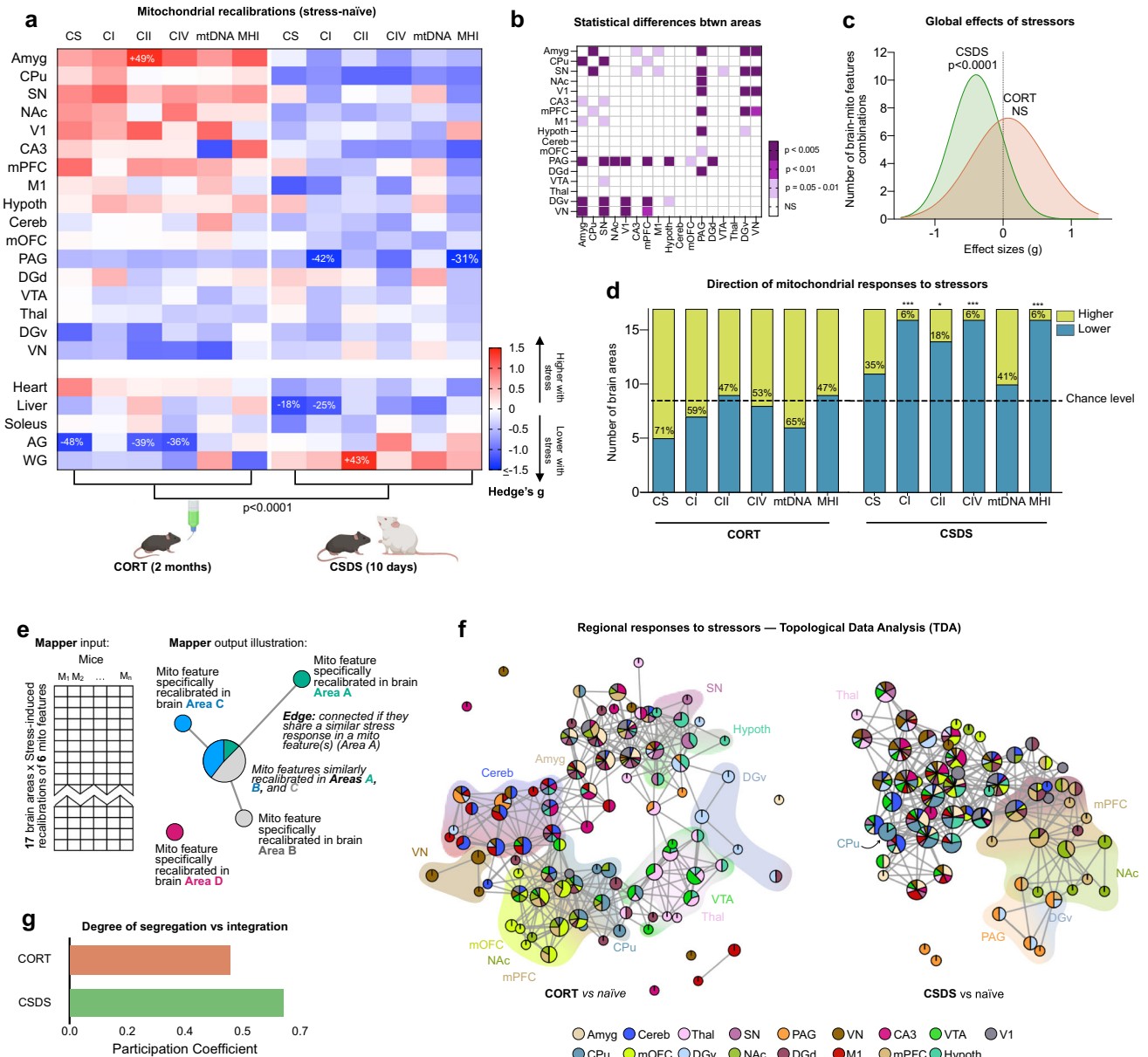

**Fig. 1 | Behavioral and neuroendocrine stressors enhance the diversity of mitochondrial phenotypes across brain areas. a** Effect of CORT and CSDS on mitochondrial features across brain areas and peripheral tissues relative to naïve mice. Effect sizes are quantified as Hedge's g, with significant effect sizes (95% confidence interval) labeled with the fold difference. Unadjusted *p*-value from two-way ANOVA, *n* = 27 mice, 5–6 per group. **b** Pair-wise comparisons between each brain areas' responses to the stressors (Hedge's g) from (A) as compared to each other area, colored by *p*-value (non-adjusted for multiple comparisons). **c** Gaussian fit for the frequency distribution of the effect sizes in A on all 6 mitochondrial features in all 17 brain areas (*n* = 102 pairs); one-sample *t*-test (two-tailed) against null hypothesis g = 0. **d** Number of brain areas in which mitochondrial features are either above or below the naïve group average in CORT and CSDS mice relative to naïve mice; *p*-values from binomial test (two-tailed); CI, CIV, MHI: *p* = 0.0003, CII:

*p* = 0.013. **e** Exemplar representations of Mapper input and output. Nodes in the Mapper output represent regional mitochondrial features (rows from input matrix) that are highly similar across mice. Thus, the brain areas that undergo similar stress-induced recalibrations in specific mitochondrial features cluster together in single nodes (most similar) or interconnected nodes (moderately similar), whereas areas that undergo divergent recalibrations are not connected. The pie-chart-based annotation of graph nodes allowed us to examine the degree of co-regulation of mito-features across brain areas. **f** Topological data analysis (TDA)-based Mapper approach to determine if brain areas were co-regulated in their stress-induced mitochondrial recalibrations for the two groups. **g** Participation coefficient (PC) representing the uniformity of mitochondrial responses across all brain areas. *CORT* corticosterone, *CSDS* chronic social defeat stress. Source data are provided as a Source Data file.

Although we also were not powered to make statistical comparisons between recovery-susceptible and -resilient mice based on the social interaction test for CSDS mice, we did observe relatively large differences in effect sizes (g > 0.8) in mitochondrial recalibrations between the groups that could be explored in the future work (Supplementary Fig. 7). Additionally, compared to non-recovered CSDS mice, animals who were allowed to recover from CSDS for 57 days

exhibited mitochondrial phenotype changes only marginally different from control mice, and only in a limited number of brain areas (Supplementary Fig. 8). This result suggests, along with the main group differences, that the brain mitochondrial recalibrations are dynamic, taking place over time scales ranging from days to weeks.

Overall, these univariate and TDA-based results established the existence of naturally occurring and acquired variation in brain

mitochondrial phenotypes between animals, providing a strong basis to test the existence of conserved associations with behaviors.

## Diversity of anxiety and depressive like behaviors

Similar to the spectrum of mitochondrial phenotypes across mice, as expected from previous work, animals also naturally exhibited large variation in their behavioral phenotypes. Behavioral tests included potential threat ("anxiety") monitored by the open-field test (OFT) and elevated plus maze (EPM), hyponeophagia monitored by novelty suppressed feeding test (NSF), and approach-avoidance conflict using the social interaction test (SI). As specific behavioral tests are generally administered in conjunction with specific interventions (CORT, CSDS), some animals were only tested on some and not all behaviors. Both CORT and CSDS produced the expected elevation in anxiety-related behavior compared to naïve mice (Supplementary Fig. 9). Results from the behavioral tests either correlated moderately with each other (EPM and OFT, $r = 0.60$), were not correlated (OFT and NSF, $r = -0.02$), or were negatively correlated (OFT and SI, $r = -0.46$). This indicated that each test captures different aspects of behavior, thereby providing a basis to examine how different aspects of behavior might relate to area-specific brain mitochondrial phenotypes.

## Brain MHI correlates with specific behaviors

We next evaluated the extent to which mitochondrial phenotypes in different brain areas were associated with behavior across all animals (Fig. 2a). Behavioral scores were transformed so that higher scores on

each test indicate higher avoidance/anxiety-like behaviors (as in ref. 24) (See *Methods*, Supplementary Fig. 9). Behavioral scores were then correlated with the 6 measures of mitochondrial phenotypes, where higher values indicate higher mitochondrial content or RC functioning. A frequency distribution of the effect sizes (Spearman r) for all mitochondrial-behavior pairs revealed a significant non-zero correlation between MHI in the 17 brain areas and behavior on OFT ($p < 0.01$), EPM ($p < 0.0001$), and SI ($p < 0.0001$), but not for NSF (Fig. 2b, Gaussian frequency distributions for the other mitochondrial features are shown in Supplementary Fig. 10). Time to feed on the NSF was capped at 600 seconds, so the correlations are less precise than for other tests.

To better understand which brain areas were driving the direction and magnitude in the distributions, we then examined the patterns of correlations for all 17 brain areas, for all 6 mitochondrial features, across the 4 behavioral tests (Fig. 2c). In the majority of brain areas, higher mitochondrial metrics were correlated with higher anxiety scores based on OFT (average $r = 0.12$) and EPM (average $r = 34$), although these overall correlations did not reach statistical significance. The strongest correlations were of $r = 0.51$ for CII in the primary motor cortex (M1) and OFT ($p = 0.025$, linear regression, unadjusted $p$-value), and $r = 0.92$ for MHI in the nucleus accumbens (NAc) and EPM ($p = 0.0005$, unadjusted $p$-value). Previous research in rodents showed that mitochondrial RC function in brain areas such as the NAc is linked to complex behaviors such as social dominance and anxiety[4,31]. For NSF there was more heterogeneity between brain areas,

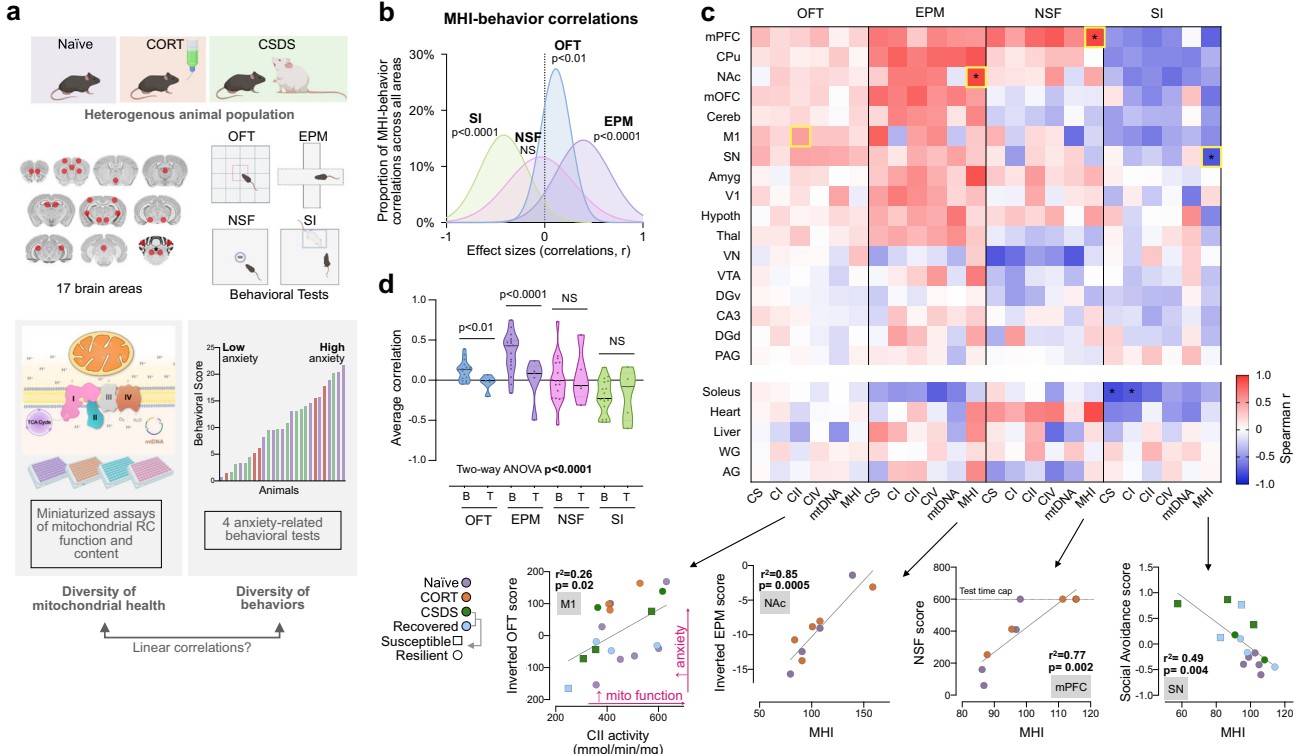

**Fig. 2 | Association patterns for brain-wide mitochondrial phenotypes and mouse behavior. a** Mitochondrial phenotyping and behavioral profiling of inter-individual variation in a heterogenous population of mice; OFT, open-field test; EPM, elevated plus maze; NSF, novelty suppressed feeding; SI, social interaction test. **b** Gaussian fits of the frequency distributions for all correlations ($n = 102$) between the composite mitochondrial health index (MHI) and each behavioral test; one-sample *t*-test (two-tailed) against null hypothesis $r = 0$ (other mitochondrial features are shown in Supplementary Fig. 8), SI: $p < 0.0001$, NSF: $p = 0.49$, OFT: $p = 0.008$, EPM: $p < 0.0001$. **c** Individual correlations for the 17 brain areas, across the 6 mitochondrial features, for each behavioral score, quantified as Spearman's r.

OFT and EPM behavioral scores were inverted so that higher scores on all four tests indicate higher anxiety (see Supplementary Fig. 8 for additional details). The strongest correlations for each behavioral test are denoted by yellow boxes, with the scatterplots shown below. An adjusted two-tailed *p*-value of <0.002 was applied (false-discovery rate 1%). All tests have been adjusted so that a higher score indicates higher anxiety-like behavior (see *Methods* for details). **d** Average correlation of each behavior for the brain (B) and tissue (T) mitochondrial features ($n = 17$ brain regions, $n = 5$ tissues); two-way ANOVA with Tukey's multiple comparison adjustment, OFT: $p = 0.0038$, EPM: $p < 0.0001$. $n = 10-27$ mice per behavioral test. Source data are provided as a Source Data file.

with a similar number of positive and negative correlations. Finally, for SI, we observed the opposite relationship; animals with higher mitochondrial content and RC activities generally displayed greater sociability (lower social avoidance) (average $r = -0.21$, strongest for MHI in the substantia nigra (SN) and SI, $r = -0.78$, $p = 0.0035$, unadjusted $p$-value). Interestingly, of the six mitochondrial features, the strongest mito-behavior correlations (for 3 out of 4 behaviors) were for MHI, which may reflect the superiority of MHI as an integrative measure of mitochondrial energy transformation capacity over individual enzymatic and molecular features, as previously observed[30,55,56].

As expected, the average correlation between RC activities and behaviors was significantly more consistent for brain mitochondria than for mitochondria in peripheral tissues (Fig. 2d). For example, whereas mitochondrial phenotypes in several brain areas correlate with anxiety-related behavior on the EPM, mitochondrial measures in the muscles, heart, liver, or adrenal glands of the same animals on average, did not correlate with behavior. This finding aligns with recent work in mice showing that mitochondrial phenotypes exhibit strong segregation between different cell types and tissues[57] and has two implications. First, it reinforces the specificity of these mito-behavior findings for the brain. Second, it implies that mitochondria across the brain and other tissues within an individual mouse are not equivalent, and likely differentially regulated[57]. This naturally raised the question whether specific brain areas within an animal could also exhibit independently regulated mitochondrial properties, and whether brain areas could be functionally organized into separate networks based on their mitochondrial properties.

## Mitochondrial phenotype-based organization of the brain

To address this question, we first asked whether mitochondrial features in each brain area/tissues were statistically independent or correlated with other tissues. Using the same rationale that underlie functional connectivity analysis in fMRI data[58–60], we first generated a correlation-based similarity (or connectivity) matrix of all mitochondrial measures across brain areas and non-brain tissues (Fig. 3a). Within the 17 brain areas, mitochondrial features were generally positively correlated (average $r = 0.22$, $p < 1^{-100}$, two-sample $t$-test), with a few exceptions. Thus, within the brain, higher mitochondrial activities in one area generally also implies higher activities in other areas ($p < 1^{-82}$, one-sample $t$-test), although the effect size is relatively modest.

Consistent with the co-regulation of RC enzymes within the mitochondrion, a modular structure was also apparent, indicating that mitochondrial features within each area were more similar than with other areas ($p < 0.0001$, permutation test). Figure 3b shows the average correlations of each mitochondrial feature with other features among all brain areas – RC enzymes were most strongly co-regulated (average rs = ~0.7–0.8), followed by mtDNA density ($r = \sim0.6$), and MHI ($r = \sim0.5$), again suggesting that MHI captures different information than individual measures. To determine how similar each area is to other areas, we computed nodal degree (i.e., average inter-regional correlation, similar to the concept of global connectivity) for all areas. Nodal degree was highest in the cerebellum (average $r = 0.34$) and lowest in the brainstem vestibular nucleus ($r = 0.05$) (Fig. 3c), suggesting that the area-to-area similarity in mitochondrial phenotypes is likely not randomly distributed across the mouse brain.

To estimate how much the observed variance of mitochondrial features across brain areas may be driven by variations in cell type composition, we correlated the abundance and proportion of various cell types including neuronal and glial cell subtypes (available from the *Blue Brain Cell Atlas*[61,62]), with the average value of each mitochondrial feature (across all animals in our cohort). Of the brain areas sampled in this study, those with more astrocytes and oligodendrocytes had higher mitochondrial enzyme activities (Supplementary Fig. 11). However, on average MHI was not correlated with any of the cell subtypes, indicating that in the mouse brain the variation in mitochondrial phenotypes between areas is only partially driven by differences in cellular composition.

## Mitochondrial enzymes-based networks

As suggested above from the divergence in the mito-behavior correlation patterns between brain areas and peripheral tissues, the mitochondrial features in peripheral tissues were not correlated with brain mitochondria (average $r = 0.02$) or with other peripheral tissues mitochondria (average $r = -0.03$) (Fig. 3d). The lack of association between brain and non-brain tissues of the same animals suggested that MHI (as well as content and RC activities) is not a ubiquitous animal-level feature, but rather relatively independently defined in the brain, and further specified among each brain area.

Given the overall positive connectivity across the brain and recent evidence of circuit-level metabolic coupling across large neural networks in the Drosophila brain[12], we examined whether certain areas exhibited particularly strong mitochondrial feature co-regulation. Functional brain networks in the living brain are defined by synchronous activity patterns (e.g., default-mode network, fronto-parietal network). Similarly, we reasoned that connectivity patterns of mitochondrial features (based on single measurements) could reflect networks of brain areas with similar bioenergetic properties. To examine this hypothesis with a framework agnostic to anatomical categorization and inclusive of all mitochondrial features, we performed multi-slice community detection analysis[63], with mitochondrial features represented in six separate layers (Fig. 3e). Categorical multi-slice community detection allows to detect cohesive groups of brain areas, or communities, that: (i) are more similar to each other than they are to the rest of the areas, and (ii) have cohesion converging across the six layers of mitochondrial features. Representing all 17 brain areas, and across 6 mitochondrial features, we found 3 separate communities, or networks, which we tentatively name: (1) *Cortico-striatal network 1*: CPu, visual, motor, mOFC, mPFC and NAc; (2) *Salience/Spatial navigation network 2*: cerebellum, vestibular nucleus, VTA, thalamus, hippocampus (CA3), and dentate gyrus (dorsal and ventral); and (3) *Threat response network 3*: amygdala, hypothalamus, substantia nigra, and periaqueductal gray (Fig. 3f).

To examine whether this functional organization of the brain revealed using biochemical mitochondrial phenotypes was also evident cross-modally, we used two Allen Mouse Brain Atlas datasets of brain-wide gene co-expression[64], and EYFP-labeled axonal projections that define the structural connectome[65], developed in the same mouse strain. Specifically, we examined whether gene expression correlations and structural connectivity within brain areas that are functionally grouped together as a network, based on mitochondrial features, is higher than expected by chance. We used two independent graph-theoretical metrics: strength fraction (S.F.)[66] and quality of modularity (Q_mod)[67], to examine whether similar communities exist in gene and structural connectivity data as in our mitochondrial phenotype data. Using permutation testing, we randomly shuffled community structure of brain areas 10,000 times to determine whether the community structure derived from mitochondrial features is also evident in gene co-expression and structural connectivity data. Both strength fraction and quality modularity statistics indicated that mitochondria-derived networks also have higher similarity in gene co-expression (S.F. $p = 0.020$; Q_mod $p = 0.008$) and structural connectome data (S.F. $p = 0.029$; Q_mod $p = 0.015$) than expected by chance (Supplementary Fig. 12). Hence, this provides convergent multimodal evidence of mitochondrial phenotypic organization overlapping with gene expression and structural connectivity.

## Network-level mitochondria-behavior correlation

To examine the potential significance and added value of this effective brain-wide mitochondrial connectivity in relation to animal behavior, we used the mitochondria-derived network results to partition the

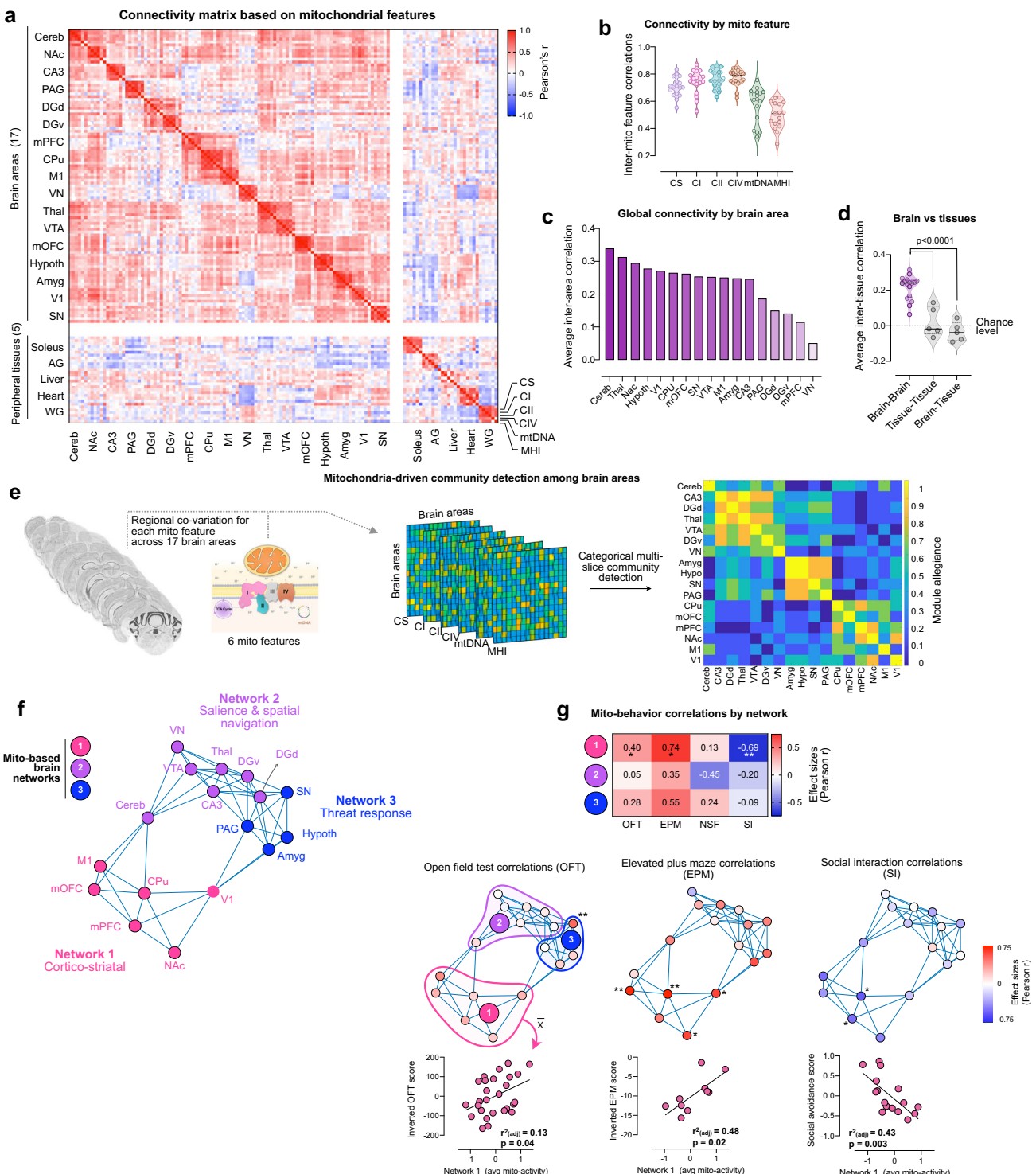

**Fig. 3 | Mitochondrial phenotype-based connectivity analysis across anatomical areas identifies large-scale brain networks that account for inter-individual variation in behaviors. a** Connectivity matrix of mitochondrial features across brain areas, using all 6 mitochondrial features across the animal cohort (n = 27 mice), quantified as Pearson's *r*. The matrix is ordered by hierarchical clustering (Euclidian distance, Ward's clustering). **b** Cross-correlation of each mitochondrial feature to the other 5 measures within each brain area (*n* = 17 brain areas). **c** Global connectivity based on the average correlation for each brain area with all other areas. **d** Average correlation of mitochondrial features between brain area's, between peripheral tissues, and between brain areas and tissues; *p* < 0.0001, Ordinary one-way ANOVA with Tukey's multiple comparisons. **e** Multi-slice community detection analysis on mitochondrial measures across the 17 brain areas (brain images acquired from the Allan Mouse Brain Atlas (Dong, H. W. The Allen reference atlas: A digital color brain atlas of the C57Bl/6 J male mouse. *John Wiley & Sons Inc.* (2008)), with mitochondrial features represented in six separate layers, resulting in **f** three distinct communities or brain networks. Modular structure confirmed by permutation test, *p* < 0.0001). **g** Average mito-behavior correlation by network for each behavioral test (top), with network 1 correlations *p*-values as follows: OFT: *p* = 0.039, EPM: *p* = 0.015, SI: *p* = 0.0033. The middle panel shows networks with each area color-coded by its average correlation with behaviors; for OFT (left), EPM (middle) and SI (right), *\*p* < 0.05, *\*\*p* < 0.01, two-tailed. The bottom panel shows scatterplots for network 1 correlations. The comparison of the modularity metrics with the modularity derived from whole-brain transcriptome and the structural connectome data, which show significant agreement, are shown in Supplementary Fig. 12. Source data are provided as a Source Data file.

brain into three networks, and then analyzed the ability of mitochondrial features in each network to linearly predict behaviors. The mitochondrial-behavior correlations varied both in strength and direction between the three networks, and the mitochondrial phenotypes among the three networks had largely divergent associations with behaviors. The average of all mitochondrial features in network 1 consistently showed the strongest average correlations with behaviors measured in the OFT, EPM, and SI tests ($r = 0.40$, $0.74$, $-0.69$; $p = 0.039$, $0.015$, $0.003$, respectively) (Fig. 3g). For the EPM, mitochondrial features in network 1 accounted for ~50% of the animal-to-animal variance in anxiety-related behavior ($0.74^2 = 0.54$). This is relatively large given the known measurement variability for behavioral testing and to a lesser extent for mitochondrial assays, and also relative to human studies[68]; however, these values are likely inflated due to the small sample size ($r^2 = 0.54$ adjusted for sample size=0.48). For the SI task, mitochondrial features accounted for 43% (adjusted $r^2$) of the inter-individual variation in social avoidance, where animals with higher Network 1 mitochondrial activities were less avoidant (Fig. 3g, *right panel*). In contrast, network 2 exhibited the weakest average correlations for the same three behaviors ($r = 0.05$, $0.35$, $-0.20$), accounting for a smaller proportion of the variance in behavior (see Fig. 3g). Together, these results support the specificity and functional significance of brain-wide mitochondrial networks, embedded within existing neural circuitry in the mouse brain.

### Validation of brain networks by transcriptional mitochondrial phenotypes

Having established that the examined 17 brain areas empirically cluster as networks based on their mitochondrial enzyme activities and mtDNA features, we then sought to (i) examine the molecular specificity of the most behaviorally relevant network 1, and (ii) to test whether a different data modality (gene expression), among a different animal cohort of the same strain, could provide converging evidence that the network 1 mitochondrial phenotype qualitatively and quantitatively differ from that of other brain areas.

Therefore, we integrated the *Allen Mouse Brain Atlas* gene expression data for each area, averaged by network (see *Materials and Methods*; $n = 16$ areas, dorsal and ventral DG are combined in the reference dataset). We then identified genes that were on average over- or under-expressed by at least a factor of 2 (double, or half) in network 1 areas relative to all other areas (Networks 2 + 3). Relative to other brain areas, network 1 was significantly enriched for processes related to synaptic signaling and transmission, neuronal and synaptic morphogenesis, and enzyme regulation by phosphorylation (Fig. 4a). In contrast, under-expressed network 1 genes were involved in metabolic processes, oxygen and chemical sensing, and anion membrane transport were. Networks 2 and 3 showed remarkably orthogonal gene expression signatures highlighting upregulation of intracellular calcium regulation, extracellular matrix organization, and response to hormonal signaling, among others genetic pathways (Supplementary Fig. 13).

Based on evidence that mitochondria functionally and molecularly specialize between tissues and cell types[14–19], we then performed a similar analysis restricted to *mitochondrial genes* only. For this, we used MitoCarta 3.0, an inventory of genes that encode proteins localized in mitochondria[16]. The resulting mitochondrial gene expression signatures for each brain area projected on a 3D principal component analysis (PCA) revealed remarkably diverse molecular mitochondrial phenotypes – or *mitotypes*[32], where all cortico-striatal network 1 areas clustered together without overlap with network 2/3 areas (Fig. 4b). This means that different brain areas express relatively unique *mitochondrial* molecular programs, which are relatively homogenous or shared among all network 1 areas.

We further examined which mitochondrial pathways distinguished each brain area by computing mitochondrial pathway scores, using gene-to-pathway annotations from MitoCarta3.0[16]. Pathway-level analysis corroborated our biochemical findings in two ways. First, network 1 areas shared similar mitochondrial phenotypic signatures (i.e., tended to cluster together in unsupervised analysis, right of heatmap). Second, as in our genome-wide analysis, several metabolism-related and energy transformation mitochondrial pathways were under-expressed among network 1 brain areas (Fig. 4c). In contrast, we found that the vestibular nucleus, a brainstem area that preferentially degenerates in mouse models of complex I defects (whereas other cortical and sub-cortical areas appear spared)[69,70], expressed the highest level of OxPhos components among all brain areas analyzed (Fig. 4c, left column). This finding possibly provides a heretofore missing molecular basis for the preferential vulnerability of the brainstem to mitochondrial OxPhos defects, but more generally highlighted the diversity and specialization of mitochondrial phenotypes across the mouse brain.

### Mitochondrial phenotypes (mitotypes) differ across brain areas

Finally, we asked which biological functions differed, and by how much, between network 1 mitochondria vs. other brain areas. We ranked the fold differences of mitochondrial pathway scores between network 1 vs. network 2 + 3, from the lowest to highest (Fig. 4d). Effect sizes were small, in part due to the substantial variation among the heterogenous network 2 + 3 areas (which cover the whole brain) that are averaged for these analyses. Network 1 mitochondria were specialized or enriched for the glycerol-phosphate (G3P) shuttle, which transfers cytoplasmic reducing equivalents (NADH) to the quinone (Q) pool to by-pass complex I and directly feed the mitochondrial respiratory chain[71], amidoxime metabolism (molybdenum-containing enzymes that reduce *N*-oxygenated structures[72]), and vitamin D metabolism. In contrast, network 1 mitochondria expressed the lowest levels of Vitamin B2 and B6 metabolism enzymes, tetrahydrobiopterin synthesis (BH4, a cofactor in the production of serotonin, dopamine, and nitric oxide[73]), and the cristae-organizing complex MICOS that physically interacts with RC complexes to enhance OxPhos capacity[74]. The direction and magnitude of differences between brain areas and networks is illustrated in bivariate mitotype plots (Fig. 4e, f) and calculated ratios of pathway scores (Fig. 4g, h), illustrating the extent to which mitochondria specialize for specific biological functions among the mouse brain.

These mitochondrial pathway-level analyses in the reference Allen Mouse Brain Atlas dataset indicate that network 1 brain areas exhibit notable molecular divergences from network 2 + 3. This result in part confirms the divergences identified in biochemical mitochondrial phenotypes, which was the basis to define network 1 in the first place. Thus, this independent analysis of brain molecular mitochondrial phenotypes in a different animal cohort further supports the existence of the behaviorally relevant mito-based networks in Fig. 3.

## Discussion

Using a high-throughput approach to functionally phenotype hundreds of brain samples from a heterogenous mouse cohort, we have defined brain-wide associations between mitochondrial phenotypes and behaviors. Combined with previous findings[31,35], the diverging mito-behavior associations between brain areas, and between brain and non-brain tissues, brought to light the possibility that different brain areas might exhibit different mitochondrial phenotypes. In particular, the network characteristics of both functional (MHI) and transcriptional mitochondrial phenotypes across the brain provided independent, converging evidence for the modular specialization of mitochondria across cortical and sub-cortical areas, as well as their relevance to animal behaviors. Based on these data, we conclude that mouse brain mitochondria may exist as behaviorally relevant networks overlapping with, but distinct from, other modalities including gene expression and structural connectivity.

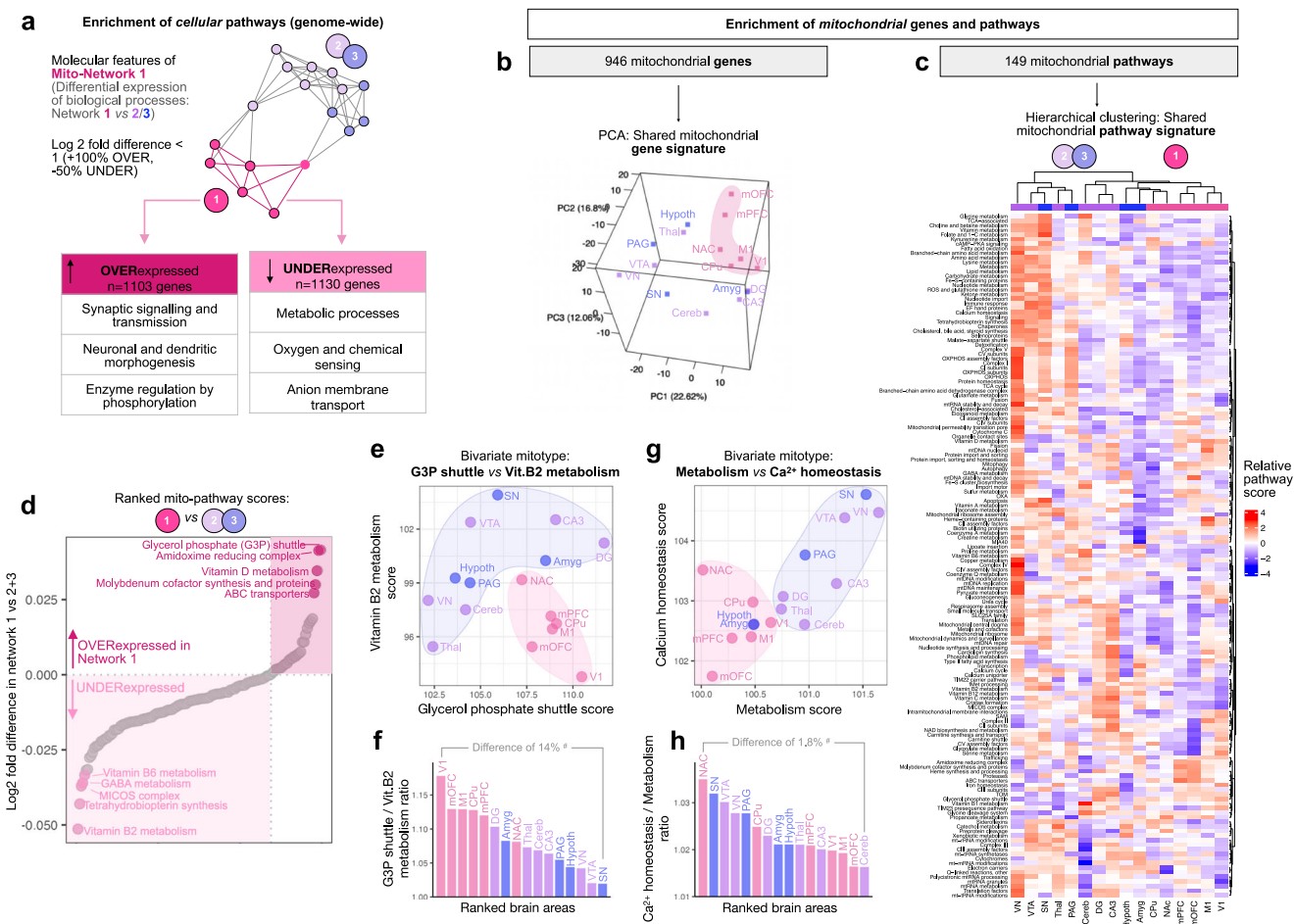

**Fig. 4 | Brain networks exhibit transcriptional genome-wide and mitochondrial specialization. a** The Allen Mouse Brain Atlas gene expression data was used to identify OVER-expressed and UNDER-expressed biological processes among Network 1 areas compared to Networks 2 + 3 areas combined. The number of genes whose Network 1 expression are more than double (Log2 fold difference >1) or less than half (Log2 fold difference <) relative to Networks 2 + 3 is listed in the tables (*right*). The top three corresponding enriched categories of biological pathways for Network 1 are listed in the tables; enriched categories for Networks 2 and 3 are available in Supplementary Fig. 13, and the gene lists are available in Supplementary File 2. **b** Principal component analysis (PCA) representation of gene expression signatures based on mitochondrial localized genes alone (*n* = 946) and **c** the expression of 149 mitochondrial pathways by brain area (MitoCarta3.0[16]),

representing the relative expression of pathways relative to all brain areas. Analyses are performed on 16 areas because gene expression for dorsal and ventral DG is combined in the reference dataset. **d** Mitochondrial pathway scores ranked by their differential expression among networks 1 vs. 2 + 3. **e**, **g** Raw scores for selected divergent mitochondrial pathways illustrating the specialization of mitochondrial phenotypes (mitotypes) between brain areas, color coded by network. **f**, **h** Computed ratios of the two pathways analyzed in e and g, quantifying the magnitude of molecular specialization between brain areas in percentage of gene expression between the two juxtaposed ratios. Ratios are derived from scaled in situ RNA hybridization data so may not represent absolute differences in transcript abundance. Source data are provided as a Source Data file.

---

Leveraging CORT and CSDS stress paradigms to induce further behavioral variation in our animal cohort also allowed us to compare the effects of these two interventions on mitochondrial phenotypes. Some brain areas were found to respond, in some cases, in opposite directions, particularly after exposure to CORT. In contrast to CORT, the recalibrations of brain mitochondria to CSDS was more uniform, with the majority of brain areas exhibiting a coordinated reduction in most mitochondrial features. This difference in mitochondrial recalibrations between both stress models may be driven by several factors. This includes the stressor duration, although the temporal dynamics over which stress-induced mitochondrial recalibrations take place remain poorly defined and will require further focused attention. Differences in the effects of CORT vs. CSDS on mitochondria also could be related to their neuroendocrine underpinnings (single hormone for CORT vs. multiple physiological neuroendocrine recalibrations for CSDS), regional differences in glucocorticoid and mineralocorticoid receptor density, or other factors generally relevant to interpreting chronic stress rodent models.

One valuable aspect of our study is that it highlights the specificity of earlier findings indicating a connection between mitochondrial RC function in the NAc and anxiety behaviors[4,31]. In our study, the strongest correlation between EPM-based anxiety-like behavior and MHI was in the NAc, confirming the strong association between NAc mitochondrial energy production capacity and anxious behavior. We also extend this finding to show that mitochondrial phenotypes are linked to behaviors across not only isolated areas, but likely distributed among brain networks. Similar to the conceptual shift from regional and cellular perspectives towards distributed brain networks, circuits, and neuronal ensembles[75,76], our findings therefore advance the notion that mitochondria may modulate brain function and behavior through distributed mito-networks. This notion is consistent with recent evidence of metabolic coupling with distributed brain-wide patterns of neural activity linked to behavior in Drosophila[12].

While we cannot directly explain why distinct mitochondrial phenotypes appear to exist across brain areas, this may be driven by three main factors. First, mitochondria could respond to differences in

neuronal circuit functioning, in agreement with observed coupling of neuronal and metabolic activities[12], such that the cellular infrastructure of neural circuits that fire together not only wire together, but also generate similar mitochondrial phenotypes (i.e., mitotypes). A second possibility is that brain areas which are regularly co-activated, i.e., within the same functional networks, harbor similar levels of receptors for neuroendocrine factors (stress and sex steroids) known to influence mitochondrial biogenesis and/or functional specialization[77,78]. A third possibility would involve differences in metabolites and substrates that regulate mitochondrial RC activities, arising similarly among co-activated brain areas. These and other potential factors underlying the modularity of mitochondria within the mouse brain remain to be investigated.

From a mitochondrial biology perspective, our data highlights a potentially important distinction between mitochondrial *content* (abundance of mitochondria) and RC *activity* normalized to mitochondrial content, such as the MHI (respiratory chain activity per mitochondrion). Frozen tissue measurements naturally reflect the maximal functional capacity of the mitochondria RC, rather than their actual in vivo rates that are driven by neural activity and metabolic demands. Interestingly, the behavioral correlations with mitochondrial content features (mtDNA density and CS activity) for social avoidance behavior were similar to one another, and differed somewhat from RC enzyme activities. Experimentally, the specificity of these findings contrasting content from RC enzyme activities highlights the value of parallel assessments of multiple mitochondrial features reflecting unique mitochondrial phenotypes (i.e., mitochondria are not all created equal). In neuroimaging terms, the composite MHI can be understood as the mitochondrial analog to fMRI-based multi-variate pattern analysis (MVPA)[79,80], where multiple features (mitochondrial enzymes for MHI, voxels for MVPA) are combined to create a more stable and statistically accurate metric of the desired outcome (mitochondrial health for MHI, or brain activation for MVPA). Moreover, mtDNAcn has previously been assessed across multiple brain areas[81], but our results go beyond these observations in showing that quantifying mtDNA content on a per-cell basis (mtDNA:nDNA ratio) is heavily skewed by cellularity variations across brain areas and not directly related to RC energy production capacity. As such, the mtDNA:nDNA ratio (mtDNAcn) is driven by how many cell bodies are present in the tissue, and correlates poorly with either mitochondrial content or RC activity. Therefore, our data reinforce the notion that mtDNAcn on its own is not a valid measure of mitochondrial phenotypes[82–84]. In the mouse brain, our findings suggest that mtDNA density per unit of volume (rather than per cell) is a more biologically meaningful mitochondrial feature when comparing brain areas that differ in cellularity.

By examining mitochondrial features across the mouse brain, we discovered a moderate level of global functional connectivity across mitochondria in most brain areas, but not among peripheral tissues (Fig. 3). By functional connectivity, we do not imply that mitochondria are directly connected to each other in the same way that neurons project and chemically (de)activate each other; but rather that they share functional properties. If mitochondrial phenotypes are directly determined by genetic and physiological factors, the logical expectation is that all mitochondria within different organs of the same organism should exhibit a high degree of coherence (i.e., correlated with each other). In other words, the animal with the highest mitochondrial content or RC activities in the one brain area should also be the animal with the highest activities in other brain areas and tissues. Our results strongly disprove this point. Similarly, previous work on multiple human tissues showed that mtDNAcn was not significantly correlated between organs, including across three brain areas[85]. Here we extend these data to 17 brain areas and 5 non-brain tissues, demonstrating that identified correlations are relatively modest (the strongest is $r = 0.31$ for the cerebellum, representing less than 10% of shared variance). Notably, while on average we observed some degree

of mitochondrial phenotype co-regulation across the brain, some brain areas exhibited *no consistent correlation* with other areas. The most parsimonious explanation for this result is that individual animals differentially recruit different circuitry, which secondarily shape their mitochondria and drive animal-specific patterns of regional mitochondrial variation. Considering theories that stipulate that the brain strives for maximal energetic efficiency[1] and that there may be limits to cellular energy conversion rates[86], we also cannot exclude the possibility that a limited quantity of resources (i.e., mitochondrial content or energy conversion) is available within the brain or the whole organism, which are distributed *unequally* in an activity-dependent manner among different organ systems, functional networks, and individual brain areas. Thus, we speculate that different animals may achieve an optimal balance of systemic and neural functions through specific combinations of mitochondrial activity in different brain areas, a possibility that remains to be tested.

In trying to better understand why certain areas displayed stronger brain-wide connectivity than others, we identified mitochondria-based communities, or networks of brain areas. Interestingly, these mitochondria-derived networks share general anatomical features with established large-scale networks. For example, the identified cortico-striatal network 1 includes the CPu, NAc, mOFC, mPFC, and motor and visual cortex, which are implicated in decision making and executing actions[87]. Network 2 (Cereb, VN, VTA, Thal, CA3, DGv, and DGd) is the most heterogenous but comprised of areas that are known to be connected and involved in salience and spatial navigation[88]. Lastly, network 3, which overlapped substantially with network 2 on several mitochondrial metrics, includes limbic and limbic-associated areas (Amyg, Hypoth, PAG and SN) involved in threat responses[89,90]. In comparing these mitochondria-derived networks to gene expression and structural connectome data we found that the communities significantly overlap across modalities. However, they are not identical, suggesting that mitochondrial properties are not the simple product of either gene expression nor structural connectivity. Finally, each network's integrated mitochondrial phenotype exhibited different associations with behavioral responses. Unsurprisingly, the cortico-striatal network 1 explained the greatest proportion of variance in behavior, accounting for approximately half of the variance (adjusted $r^2 \sim 43$–48%) in anxiety-related behavior between animals. Removing the intractable variance lost to measurement error intrinsic to both behavioral and mitochondrial assays (i.e., total explainable variance is less than 100%) brings the proportion of *explainable* variance accounted for by network 1 mitochondrial biology at considerably more than half. This quantitative observation increases the likelihood that the identified large-scale mitochondrial networks are functionally relevant. This mito-based network perspective may provide a basis to further delineate and develop accurate models brain-wide mitochondria organization and specialization.

In relation to the specialization of mitochondrial phenotypes across the mouse brain, the transcriptional signatures from the Allen Mouse Brain Atlas also confirmed that brain mitochondria are not all created equal. Our results thereby extend previous work highlighting distinct molecular and functional mitochondrial phenotypes in different brain cell types[14–19], and document several new observations of area-specific mitotypes (see Fig. 4c). Understanding the origin and functional significance of mitotype variations, as well as the extent to which such mitotype variation applies to other species, including in the human brain, are exciting questions for future research.

Until now, the methods used to reveal neurobiological and metabolic networks in mammals have typically been through indirect functional and structural connectivity analysis[20]. Here, we have developed a scalable approach to examine mitochondrial phenotypes across a large number of brain areas in mice with a range of behavioral phenotypes. We showed that mitochondrial phenotype connectivity was non-random, and linked with gene co-expression and structural

connectivity, thereby providing converging evidence of mitochondria-based networks across modalities. This study synergizes with recent work[12] providing the technical and empirical foundation to bring mitochondrial biology into brain-wide, network-based models of neural systems in mammals. Applied to mitochondria, network-based analytics should contribute to develop more accurate maps, and eventually an understanding of what drives mitochondrial and metabolic properties across interacting neural circuits. Developing a spatially resolved understanding of brain mitochondrial biology will help to resolve the energetic constraints on brain function and behavior.

### Limitations

Notable limitations of this study include the lack of cell type specificity. Neurons operate in a metabolic partnership with astrocytes and glial cells[91], and different cell types exhibit different molecular mitochondrial phenotypes (e.g., ref. 92,) that cannot be disentangled in tissue homogenates. While neither enzymatic nor functional mitochondrial profiling at the single brain cell is currently technically feasible, it remains possible that mitochondrial phenotypes between brain areas are influenced by differences in cell type proportions. Moreover, our molecular and biochemical mitochondrial phenotypes do not reflect other factors that can influence the efficiency or activity of the mitochondrial RC or OxPhos system in vivo, such as variations in cofactor abundance and RC structural assembly (e.g., supercomplexes)[93]. As functional assays require harvesting brain tissue, it also was not feasible to ascertain within a given animal how stable (trait) or dynamic (state) the brain biochemical mitochondrial phenotypes are. If mitochondrial phenotypes were more dynamic than expected, our estimated proportion of behavioral variance attributable to mitochondrial biology could be substantially underestimated.

## Methods

### Animals

This study was carried out in accordance with NIH Guidelines and was approved by the Institutional Animal Care and Use Committee (IACUC) at New York State Psychiatric Institute, protocol number NYSPI-1545. Adult (52 weeks old) *C57BL/6J* male mice were obtained from Jackson Laboratories ($n = 29$). Only one sex was used to keep sample size technically feasible, while enabling analyses on the largest possible number of tissues and animals. Males were shamefully selected because there is more literature available for comparison. Three month-old *CD1* retired breeder male mice were obtained from Charles River, and were used as the aggressors in the social defeat model. Mice were housed in a 12-h light-dark colony room, with food and water provided ad libitum. Behavioral testing was performed during the light phase. *C57BL/6J* mice were pair-house while CD1 aggressors were single housed.

### Chronic social defeat stress (CSDS)

**Aggressors prescreening.** All *CD1* mice used in the experiment were pre-screened for aggressive behaviors as previously described[50]. During a three-day screening procedure, a novel *C57BL/6J* mouse was placed in the cage of the *CD1* mouse for 3 min. *C57BL/6J* screener mice were not further used in the study. The latency of the *CD1* mouse to attack the *C57BL/6J* screener mouse was recorded. *CD1* mice that attacked in less than 1 min on at least the last two consecutive screening days were considered to be aggressive.

**Experimental groups.** Thirteen *C57BL/6J* mice underwent social defeat stress, while 6 remained in their cages, to serve as the naïve group. Animals were randomly assigned to these groups. One mouse subjected to social defeat died of unknown causes during the duration of the experiment. Of the remaining 12 social defeat mice, half ($n = 6$) were sacrificed two days after the completion of the stressor, and they are referred to as the 'stressed' CSDS group. Three naïve mice were sacrificed at the same time. The other half of the CSDS mice ($n = 6$)

along with 3 naïve mice, were allowed an 8.5 weeks (59 days) stress recovery period prior to sacrifice. This group is referred to as the 'recovered' group. Animals were selected for the recovery group from the CSDS group so that there was an even ratio of susceptible to resilient mice in the 'stressed' and 'recovered' groups.

**Social defeat paradigm.** Adult experimental male *C57BL/6J* mice ($n = 13$) were exposed to a CSDS paradigm daily, for 10 days. The experimental *C57BL/6J* mice were placed in the cage of a new *CD1* aggressor mouse for 5 min every day, for 10 consecutive days. After the 5 min of physical defeat, the *C57BL/6J* mice were housed in the same cage as the *CD1* aggressor, with a perforated plexiglass divider to separate them for 24 h. After the 10 days of defeats, experimental *C57BL/6J* mice and *CD1* mice were singly housed. Adult naïve male *C57BL/6J* mice ($n = 6$) were housed 2 mice per cage separated by a perforated plexiglass divider. To control for the effects of experimental handling, each naïve mouse was paired with a new naïve mouse every day for 10 days.

### Corticosterone administration (CORT)

Corticosterone (Sigma Aldrich, St Louis, MO) was dissolved in vehicle (0.45% ß-cyclodextrin) at a concentration of 35 μg/ml, equivalent to administration of ~5 mg/kg/day per mouse[48]. *C57BL/6J* mice ($n = 5$) were group-housed and administered corticosterone in their drinking water. Naïve mice ($n = 5$) were group-housed and received only vehicle. Animals were randomly allocated to the two groups. Water bottles were prepared twice a week, and the mice never had access to other water. Behavioral testing began on day 56 of CORT administration. Experimental and naïve mice were sacrificed on day 63 following the completion of behavioral testing. The duration of administration was 9 consecutive weeks. The study design with the number of animals allocated to each condition are listed in Supplementary Table 1 and in Supplementary Fig. 2.

### Behavioral tests

**Social interaction (SI) test.** Social avoidance was measured 24 h after the completion of the last day of defeat (day 11). During the first trial, experimental mice were allowed to explore an open-field arena ($40 \times 40 \times 40$ cm) containing an empty wire enclosure for 2.5 min. During the second trial, a CD1 mouse was placed into the wire enclosure, and the C57BL/J6 mouse was reintroduced for 2.5 min. Time spent in the SI zone and time spent in the corner zones during the first and second trial were recorded. SI ratios were calculated as time spent in the SI zone during the second trial divided by time spent in the interaction zone during the first trial[24]. Mice with SI ratios of <1 were considered 'susceptible' ($n = 6$), while mice with SI ratios >1 were considered 'resilient' ($n = 7$)[49]. Corner zone ratios were calculated as time spent in the corner zones during the second trial divided by time spent in the corner zones during the first trial.

**Open-field test (OFT)- both groups.** OFT were run 24 h after the SI tests for CSDS mice, and on day 56 of CORT administration. Each mouse was placed in an open-field arena ($40 \times 40$ cm $\times 40$ cm) for 10 min. A camera on a tripod stand was set up above the arena to record the activity, and the video was later analyzed using Ethovision XT (Noldus). The percent of time spent in the center of the open field ($20$ cm $\times 20$ cm) and the percent distance traveled in the center of the open field were analyzed.

**Elevated plus maze (EPM).** EPM tests were done the day after the OFT for CORT mice (day 57 of treatment). Each mouse was placed in an elevated plus maze for 10 min. A camera on a tripod stand was set up above the arena to record the activity, and the video was later analyzed using Ethovision XT (Noldus). The percent of time spent in the open arms was analyzed.

**Novelty suppressed feeding (NSF).** NSF was performed after EPM for CORT mice (day 59 of treatment), and as previously described[94]. Briefly, NSF testing apparatus consisted of a plastic box (50 × 50 x 20 cm) with 2 cm of wood chip bedding. The center of the arena was brightly lit (1200 lux). Mice were food restricted for 15 h during the dark phase prior to testing. At the time of testing, a single pellet of food (regular chow) was placed on a white paper platform positioned in the center of the box. Each animal was placed in a corner of the box, and a stopwatch was immediately started. The latency of the mice to begin eating was recorded during a 10-min test. Immediately after mice took a bite from the food pellet, the pellet was removed from the arena. Mice were then placed back in their home cage and latency to eat and the amount of food consumed in 5 min were measured (home cage consumption). NSF latency was capped at 10 min, with animals that did not consume any portion of the pellet receiving a score of 600 s.

**Behavioral z-scores.** Social avoidance scores were determined by averaging the z-scores for 4 measures as previously described[24]; SI ratio, corner ratio, time spent in SI zone, and time spent in corner zones. The z-scores for SI ratio and time spent in SI zone were multiplied by −1, so that higher scores across all 4 measures indicated higher avoidance. Similarly, the OFT score was determined by averaging the z-scores for the 2 measures. The z-scores were multiplied by −1, so that higher OFT score indicate higher avoidance of the brightly lit center of the OF. EPM scores were also multiplied by −1, so that higher EPM scores indicate higher avoidance of the open arms of the EPM. High NSF scores already indicate higher avoidance, so they were not inverted. Therefore, across all 4 tests, higher scores indicate higher anxiety-like/avoidant behavior.

**Tissue collection**
Animals were sacrificed by rapid decapitation to maintain mitochondrial integrity in brain tissue. Brains were rapidly flash frozen in ice-cold isopentane, stored at −80 °C, and later transferred to −170 °C (liquid nitrogen vapor) until mitochondrial measures were performed. Brains were transferred back to −80 °C during the week prior to being sectioned and were then transferred to −30 °C the night before sectioning. Brains were sectioned coronally on a Leica Model CM3050 S cryostat. The cryostat internal temperature and blade temperature were set to −22 °C during sectioning. Brains were mounted onto a specimen disk using optimal cutting temperature (OCT) compound. The brain was sectioned coronally, alternating between two 200 μm thick slices and then two 20μm, and were deposited onto microscope slides. Brain sections were kept at −80 °C until brain area-specific tissue sample collection. One of the recovered CDSD mouse brain contained blood and could not be reliably sectioned, so this animal's brain tissue was excluded from analysis. A second CSDS mouse brain and a naïve brain both cracked during slicing, so some of the brain areas (n = 7 for CSDS mouse, n = 8 for naïve) could not be obtained from those brains, but the areas that were obtained were included in analysis.

**Tissue biopsy punches on frozen brain sections.** The scalable Allen Mouse Brain volumetric atlas 2012[95] was used to determine the location of each brain area of interest, and their distance from bregma. The atlas' Nissl-stained images were used as a reference for estimating each section's distance from bregma, which determined which brain slices would be used for punch biopsy collection of each area of interest (Supplementary Fig. 3b). Brain slices were punched using 1.00 mm diameter Robbins True-Cut Disposable Biopsy. Two bilateral punches were collected for each brain area over dry ice. For brain areas in the midline, punches were taken from two consecutive slices. All tissue punch locations were approximated by using landmarks on the slice and by comparing to the atlas. It is important to note that the 1 mm

puncher was larger than the actual brain area in some instances, and so parts of neighboring brain areas may have been included in the punches. Therefore, the punching technique may include some error. Punches were stored at −80 °C until they were ready to be used for enzymatic activity assays. Intact consecutive 200μm cerebellar slices were stored at −80 °C for immunohistochemical staining.

The tissue punches were too light to be accurately weighed, so we had to estimate the weight based on the reported mouse brain density of 1.04 g/cm[3,96]. The punches were 0.5 mm (diameter) by 200μm (height), so using the equations $v = \pi r^2 * h$ and $d = \frac{m}{v}$ we obtain an estimated mass of 0.163 mg per punch, thus 2 punches were approximated to weigh 0.327 mg.

**Mitochondrial measurements**
**Tissue preparation.** The punches from each brain area were homogenized in 0.2 mL of homogenization buffer (1 mM EDTA and 50 mM Triethanolamine) (2x 1 mm tissue punch/0.2 mL of homogenization buffer), with 2 tungsten beads to disrupt the tissues' cells and release the mitochondria. Tissues were homogenized using a Tissue Lyser (Qiagen cat# 85300), which was run at 30 cycles/s for 1 min. The tissues were then incubated in ice for 5 min, and were then re-homogenized for 1 min. Tissues were vortexed to ensure homogeneity. Peripheral tissues were cut over dry ice, weighed, and were then homogenized 1:180 (weight:volume, mg: μL), except for heart samples that were further diluted to 1:720 to be in the dynamic range of the assays.

**Enzymatic activities**
Enzymatic activities were quantified spectrophotometrically for Citrate Synthase (CS), complex I (CI, NADH-ubiquinone oxidoreductase), Succinate Dehydrogenase (CII, succinate-ubiquinone oxidoreductase, also known as SDH), Cytochrome C Oxidase (CIV, COX) and were expressed per mg of tissue, as described previously[32], with some modifications as described below in full details. All miniaturized assay measurements were performed in 96-well plates and enzymatic activity assays recorded on a Spectramax M2 (Spectramax Pro 6, Molecular Devices). Linear slopes reflecting changes in absorbance of the reporter dye were exported to Microsoft excel and converted into enzymatic activities using the molar extinction coefficient and dilution factor for each assay. The assays were optimized to determine the minimal amount of tissue required to obtain reliable results assessed by the C.V. between duplicates. The assays were then further optimized to determine the minimal amount of brain tissue homogenate required for each individual assay. Assay validation involved regressing increasing tissue amounts (number of punches) with observed activities, which confirmed that an increase or decrease in tissue used produced proportional changes in total activity.

To validate the miniaturized biochemical enzymatic assays, we performed each assay in both the miniaturized 200ul format in 96-well plates, and the traditional 1 mL cuvette format, using increasing tissue homogenate volumes (4, 8, 12, 16, 20 μL) from the same brain area (cerebellum) of a wild-type control mouse. The same homogenized tissue (1:200 weight:volume, mg:μL) was used for CS, CI, CII, and CIV spectrophotometric assays from which the respective enzyme activities were quantified using the reagents and procedures described below. Both the miniaturized and standard-size cuvette assays showed high agreement ($r^2 = 0.81–0.96$; ps = 0.0032–0.037) (Supplementary Fig. 1). For 3 out of 4 assays, compared to the traditional 1 mL cuvettes, the 96-well plate exhibited substantially less variation between consecutive dilutions (CS, 8% in the 96-well plate vs. 20% in the cuvette; CI, 15% vs. 13%; SDH, 4% vs. 9%; COX, 5% vs. 9%).

Samples were run in duplicates for each enzyme, along with a non-specific activity control, and every plate had a positive control (heart homogenate). The 96-well plates were designed so that each brain area/tissue from all animals were run on a single plate, which prevents potential batch variation for comparisons between the animals.

However, due to the size of the plates, no more than 2 types of tissues could be run together on a single plate. This limits the accuracy of the comparisons of enzymatic activities between tissues, but maximizes the accuracy of comparisons between animals for each tissue. Reference positive controls on each plate were used to control for potential batches/plates effects. None of the enzymatic activities of some samples ($n = 8$ out of 579, resulting in $n = 571$) could be measured for technical reasons and are therefore not included in the analyses.

Citrate synthase (CS) enzymatic activity was determined by measuring the increase in absorbance of DTNB at 412 nm at −30 °C in 200 µL of a reaction buffer (200 mM Tris, pH 7.4) containing acetyl-CoA 10 mM, 10 mM 5,5'- dithiobis- (2-nitrobenzoic acid) (DTNB), 2 mM oxaloacetic acid, and 10% w/v Triton-x-100. The rate of conversion of DTNB into $NTB^{2-}$ ions indicates the enzymatic activity and is used as a marker of mitochondrial content. Oxaloacetate is removed from the assay mix as a way to measure non-specific activity. The final CS activity was determined by integrating OD412 change over 150–400 s and then subtracting the non-specific activity. 10 µL of homogenate was used to measure CS activity.

Complex I (CI, NADH-ubiquinone oxidoreductase) activity was determined by measuring the decrease in absorbance of DCIP. The rate of absorbance of DCIP is measured at 600 nm at 30 °C, in 200 µL of a reaction buffer (potassium phosphate 100 mM, pH 7.4) containing 550 mg/mL bovine serum albumin (BSA), 50 mM potassium cyanide (KCN), 20 mM decylubiquinone, and 0.4 mM antimycin A. 10 µL of homogenate was used to measure CI activity. Antimycin A and KCN are used to inhibit electron flow through complexes III and IV. The negative control condition includes rotenone (200 mm) and piericidin A (0.2 mM), which selectively inhibit NADH-ubiquinone oxidoreductase. The final CI activity was determined by integrating OD600 change over 150–500 s, and by subtracting the rate of NADH oxidation in the presence of rotenone and piericidin A from the total decrease in absorbance.

Complex II (CII, succinate-ubiquinone oxidoreductase, also known as SDH, succinate dehydrogenase) activity was determined by measuring the decrease in absorbance of DCIP. The rate of absorbance of DCIP was measured at 600 nm at 30 °C, in 200 µL of a reaction buffer (potassium phosphate 100 mM, pH 7.4) containing 50 mg/mL bovine serum albumin (BSA), 500 µM rotenone, 500 mM succinate-tris, 50 mM potassium cyanide (KCN), 20 mM decylubiquinone, 20 mM DCIP, 50 mM ATP, 0.4 mM antimycin A. 15 µL of homogenate was used to measure CII activity. The negative control condition includes sodium-malonate, which inhibits succinate-ubiquinone oxidoreductase. The final CII activity was determined integrating OD600 change over 300–800 s, and by subtracting the absorbance in the presence of malonate (500 mM) from the total decrease in absorbance.

Complex IV (CIV, also cytochrome c oxidase) activity was determined by measuring the decrease in absorbance of cytochrome c. The rate of conversion of cytochrome c from a reduced to oxidized state was measured at 550 nm at 30 °C, in 200 µL of reaction buffer (100 mM potassium phosphate, pH 7.5) containing 10% w/v n-dodecylmaltoside and 120 µM of purified reduced cytochrome c. 6 µL of homogenate was used to measure COX activity. The negative control condition omits tissue homogenate determine the auto-oxidation of reduced cytochrome c. The final CIV activity was determined by integrating OD550 change over 150–500 s, and by subtracting the non-specific activity from the total decrease in absorbance.

Mitochondrial enzymatic activities were determined by averaging the duplicates. The technical variation of the duplicates was measured with and a 10% cutoff. The specific activity of each sample was calculated as the total activity minus non-specific activity (negative control), times the normalization factor. Plates were normalized by their positive controls. Because of some variation in positive controls, each plate's positive control was z-scored to the average of the positive control activity per assay. All of the activities on the plate were then multiplied by their normalization factor, which is determined by 1/z-scored positive control.

## Mitochondrial DNA (mtDNA) quantification

mtDNA density and mtDNA copy number (mtDNAcn) were measured as previously described[97], with minor modifications. The homogenate used for the enzymatic activity measures was lysed at a 1:10 dilution in lysis buffer (100 mM Tris HCl pH 8.5, 0.5% Tween 20, and 200 g/mL proteinase K) for 10 h at 55 °C, 10 min at 95 °C, and were kept at 4 °C until used for qPCR. qPCR reactions were measured in triplicates in 384 well qPCR plates using a liquid handling station (ep-Motion5073, Eppendorf), with 12 µL of master mix (TaqMan Universal Master mix fast, Life Technologies #4444964) and 8 µL of lysate. Each plate contained triplicates of a positive control (heart) and of a negative control (lysate without homogenate). qPCR reaction with Taqman chemistry was used to simultaneously quantify mitochondrial and nuclear amplicons in the same reactions: Cytochrome c oxidase subunit 1 (COX1, mtDNA) and β−2 microglobulin (B2M, nDNA). The Master Mix included 300 nM of primers and 100 nM probe: COX1-Fwd: ACCACCATCATTTCTCCTTCTC, COX1-Rev: CTCCTGCATGGGCTAGATTT, COX1-Probe: HEX/AAGCAGGAG/ZEN/CAGGAACAGGATGAA/3IABkFQ. mB2M-Fwd: GAGAATGGGAAGCCGAACATA, mB2M-Rev: CCGTTCTTC AGCATTTGGATTT, B2M-Probe: FAM/CGTAACACA/ZEN/GTTCCACCC GCCTC/3IABkFQ.

The plate was quickly centrifuged and cycled in a QuantStudio 7 flex instrument (Applied Biosystems Cat# 448570) at 50 °C for 2 min, 95 °C for 20 s, 95 °C for 1 min, 60 °C for 20 s for 40x cycles. To ensure comparable Ct values across plates and assays, thresholds for fluorescence detection for both mitochondrial and nuclear amplicons were set to 0.08. Triplicates for each sample were averaged for mtDNA and nDNA, and an exclusion cutoff of Cts >33 was applied. For samples with triplicates C.V.s > 0.02, the triplicates were checked, and outlier values removed where appropriate, and the remaining duplicates were used. The final cutoff was C.V. > 0.1 (10%); and any samples with a C.V. > 0.1 were discarded. The mtDNAcn was derived from the ΔCt calculated by subtracting the average mtDNA Ct from the average nDNA Ct. mtDNAcn was calculated by $2^{(\Delta Ct)} \times 2$. For measures of mtDNA density, the Ct value was linearized as $2^{Ct}/(1/10^{-12})$ to derive relative mtDNA abundance per unit of tissue.

In tissues of similar cellular density (number of cell nuclei per area or mass of tissue), mtDNAcn (mtDNA:nDNA) provides an accurate reflection of mtDNA genome density per cell. However, different brain areas vary widely in their cellularity (up to 8.5-fold), mostly because some defined areas such as the granular layer of the cerebellum are populated with numerous small cell bodies, whereas other areas such as the molecular layer of the DG are completely acellular and devoid of cell bodies/nuclei. Nevertheless, acellular tissue compartments filled with dendrites can be rich in mitochondria and mtDNA, and therefore the number of mtDNA copies per unit of tissue (µm³ or mg of tissue) is a more generalizable and accurate estimate of mtDNA density between brain areas.

## Mitochondrial Health Index

The mitochondrial health index (MHI) integrates the 5 primary mitochondrial features, yielding an overall score of mitochondrial respiratory chain activity on a per mitochondrion basis[30,32]. The simple equation uses the activities of Complexes I, II, and IV as a numerator, divided by two indirect markers of mitochondrial content, citrate synthase activity and mtDNA density: MHI = (CI + CII + CIV)/(CS+mtDNA density+1) * 100. A value of 1 is added as a third factor on the denominator to balance the equation. Values for each of the 5 features are mean centered (value of an animal relative to all other animals) such than an animal with average activity for all features will have an MHI of 100 [(1 + 1 + 1)/(1 + 1 + 1) * 100 = 100].

## Immunohistochemical staining

Immunofluorescent staining was used to quantify local Succinate dehydrogenase complex, subunit A (SDHA) expression in the cerebellum. Slides with 200 μm coronal sections bearing the cerebellum were taken from storage at −80 °C and allowed to warm up to room temperature for 1 h. The sections were obtained during tissue collection from the same brains as those used for measuring enzymatic activities, but were stored at −80 °C and were not biopsied (Supplementary Fig. 4b). The sections were then fixed in freshly prepared 4% PFA for 5 min and immediately washed twice in a 0.1% PBS-Tween 20 (PBS-Tw) solution. They then underwent dehydration through iced cold 70% methanol for 10 min, 95% methanol for 10 min, 100% methanol for 20 min, 95% methanol for 10 min and 70% methanol for 10 min followed by two washes in PBS-Tw. In a PBS-Tw bath, sections were carefully unmounted from the slides using a razor blade and artist's paintbrush. One section containing an appropriate cerebellum area was selected from each animal and individually placed in 10% Normal Donkey Serum (NDS) overnight at 4 °C. They were then incubated in a 1:100 dilution of Rabbit anti-SDHA 1° antibody in PBS overnight at 4 °C on a gentle shaker. The following day, they were washed repeatedly in PBS-Tw. Sections were then incubated in a 1:500 dilution of donkey anti-Rabbit 2° conjugated with AlexaFluor 546 in NDS for 2 h in a dark box at room temperature. They were then washed in PBS, incubated in 0.625 μg/mL 4′,6-diamidino-2-phenylindole (DAPI) and washed once more. Finally, sections were mounted on Superfrost slides with Prolong Diamond mounting media and a No. 1.5 coverslip.

## Confocal imaging

The immunofluorescent staining was imaged on a Leica SP8 confocal microscope equipped with Lightning super resolution software. The areas of interest were located using a Leica 20 × 0.75 NA objective to determine areas of suitable staining surrounding the purkinje cell layer in the cerebellum. High-resolution images of the area were taken with a Leica 63× 1.40 NA oil objective. For highest resolution, the images were formatted at 2496 × 2496 pixels, with a zoom of 2.50x, line average of 8, pinhole size of 0.50 AU. The final pixel size was 29.58 x 29.58 nm and a z-stack with a step size 0.15 μm was taken through the stained area. A 405 nm laser was used for exciting the DAPI channel with a power of 8.0% and gain of 60%. A 561 nm laser was used for the AlexaFluor 546 channel with a power of 4.0% and a gain of 20%.

## SDHA staining analysis

Final images were deconvolved in the Lightning software using the default settings. Images were analyzed in ImageJ version 1.52p. Fifty consecutive slices from each stack were selected for a section thickness of 7.5 μm, which roughly covered the depth of penetration of the antibodies. The 16 bit images were first thresholded using the Triangle algorithm within the integrated Auto-thresholding plugin to create a binary image. The particle analyzer was then used to quantify the percent of the area stained for each slice. The percent area stained from each slice was summated to calculate the percent of volume stained. For compartmentalized data, the molecular and granular layers were separated using a rectangular selection of approximately 50 px width through each stack. For the purkinje layer, a rectangle was drawn that narrowly captured the stained area of each slice. The volumetric staining for each area of interest was calculated from the percent of area stained quantified using the particle analyzer.

## TDA-based mapper analysis

After creating a delta of mitochondrial features between stressed (CORT or CSDS) and naïve mice, the data matrix for each group was processed through the TDA-based Mapper pipeline[52]. The input data matrix contained 102 concatenated rows for brain areas (17 brain areas x 6 mitochondrial features) and 5 or 6 columns for individual mice, based on the number of animals per group. Missing values, if any,

in the input data matrix were interpolated within group using a linear interpolation. The TDA-based Mapper analysis pipeline consists of four steps. First, Mapper involves embedding the high-dimensional input data into a lower dimension $d$, using a filter function $f$. For ease of visualization, we chose $d = 2$. The choice of filter function dictates what properties of the data are to be preserved in the lower dimensional space. For example, linear filter functions like classical principal component analysis (PCA) could be used to preserve the global variance of the data points in the high-dimensional space. However, a large number of studies using animal models and computational research suggest that inter-regional interactions in the brain are multi-variate and nonlinear[98–100]. Thus, to better capture the intrinsic geometry of the data, a nonlinear filter function based on neighborhood embedding was used[52]. The second step of Mapper performs overlapping n-dimensional binning to allow for compression and reducing the effect of noisy data points. Here, we divided the data into lower dimensional space into 64 bins with 70% overlap. Similar results were observed for different number of bins (e.g., 49 and 81). Third, partial clustering within each bin is performed, where the original high-dimensional information is used for coalescing (or separating) data points into nodes in the low-dimensional space. Partial clustering allows to recover the loss of information incurred due to dimensional reduction in step one[101]. Lastly, to generate a graphical representation of the shape of the data, nodes from different bins are connected if any data points are shared between them.

The Mapper-generated graphs can be annotated (or colored) using meta-information that was not used to construct the graphs. Here, we annotated these graphs using area-labels to examine whether mitochondrial features were similarity expressed across all areas or whether regional specificity was observed across the two groups. To quantify the extent of segregation (high regional specificity) or integration (low regional specificity) across the six mitochondrial features, we used a graph-theoretical measurement of participation coefficient[54]. Participation coefficient $P_i$ of a node $i$ is defined as:

$$P_i = 1 - \sum_{s=1}^{N_M} \left(\frac{\kappa_{is}}{k_i}\right)^2 \tag{1}$$

where $\kappa_{is}$ is the number of links of node $i$ to nodes in community $s$, $k_i$ is the total degree of node $i$ and $N_M$ is the number of communities. The $P_i$ of a node $i$ is close to 1 if its links are uniformly distributed among all communities of the graph (and hence integrated) and it is close to 0 if its links are mostly within its own community (and hence segregated).

## Multi-slice community detection

One of the most commonly studied mesoscale aspect of a graph is modularity, where highly modular graphs consist of cohesive groups of nodes (or communities) that are more strongly connected to each other than they are to the rest of the network[102]. Examination of modularity has been recently extended to multi-slice networks that are defined by coupling multiple adjacency matrices across time or modality[63]. Here, we used six slices, each derived from one of the mitochondrial features, where each slice represented weighted adjacency matrix of pair-wise Pearson's correlation between brain areas. We use categorical multi-slice community detection algorithm with the presence of all-to-all identity arcs between slices[63]. The generalized modularity function for multi-slice community detection is given by

$$Q_{\text{multislice}} = \frac{1}{2\mu} \sum_{ijsr} \left[ \left( A_{ijs} - \gamma_s \frac{k_{is}k_{js}}{2m_s} \right) \delta_{sr} + \delta_{ij} C_{jsr} \right] \delta(g_{is}, g_{jr}) \tag{2}$$

Where $A_{ijs}$ represents weighted adjacencies between nodes $i$ and $j$ for each slice $s$, with interslice couplings $C_{jrs}$ that connect node $j$ in slice $r$ to itself in slice $s$. $\gamma_s$ represents resolution parameter in each slice; higher value of $\gamma_s$ (e.g., >1) detects smaller modules, while lower values

(e.g., between 0 and 1) detects bigger modules. In line with previous work, here, we set the resolution parameter across slices to be unity, i.e., $\gamma_s = 1^{103}$. For simplicity, and as done previously, the interslice coupling parameters were also kept same across slices[63]. Here, we used $C_{jsr} = 0.1$. Perturbation of interslice coupling values around 0.1 produced similar results.

To estimate the stability of identified communities across the six mitochondrial slices, under the assumption that stable communities could represent convergence across mitochondrial features, we estimated module allegiance matrix[104]. The module allegiance matrix summarizes the stability of community structure across slices, such that each entry $i,j$ of the matrix corresponds to the percentage of slices in which areas $i$ and $j$ belong to the same community. Finally, an iterative consensus community detection algorithm was run on the module allegiance matrix to define the large-scale networks that are convergent across slices defined by mitochondrial features. The iterative community detection[105] was run multiple times (1000), followed by consensus clustering to get stable results for identifying large-scale networks of brain areas[106].

### Comparison of mito-based networks with transcriptomic- and structural connectivity-based networks

To compare our mitochondrial communities against other modalities, we examined the organization of gene co-expression in the mouse brain as well as mouse structural connectome data. We utilized the Allen Brain Atlas' ISH (in situ hybridization) feature, which maps each gene onto a reference standard coordinate atlas image, providing a spatial estimate of transcript levels representing gene expression[64]. Specifically, we used the Anatomic Gene Expression Atlas (AGEA) (https://mouse.brain-map.org/agea), which is a a three-dimensional male adult *C57BL/6J* mouse brain atlas based on the ISH gene expression images. The AGEA feature allows for selecting both a 'seed voxel' and a target 'selected voxel' from exact brain coordinates, yielding the transcriptome-wide correlation between the seed and target areas. The correlation is a measure of the average co-expression between the two voxels. The co-expression values were obtained for all possible pair of brain areas ($17 \times 17$ matrix), yielding a gene co-expression correlation matrix to which the structure of our mito-based networks could be compared.

The structural connectome data were obtained from The Allen Mouse Brain Connectivity Atlas[65], which is a mesoscale connectome of the adult mouse brain. We utilized the normalized projection strength values for the brain areas of interest. Because this atlas does not distinguish between dorsal and ventral dentate gyrus, data was obtained for 16 brain areas. For three brain areas where the Atlas provides connectivity values for multiple sub-areas (OFC, VN, Cerebellum), the connectivity values were averaged across the sub-areas to yield a global measure for each area, thus matching the dimensionality of our mitochondrial dataset.

We used non-parametric permutation statistics with 10,000 permutations to examine whether the mitochondrial features derived networks were also more densely connected than expected by chance in other modalities (transcriptomic and structural connectivity data). To measure the degree of within-network connectedness we used two established metrics: modularity index (Q_mod;[67]) and strength fraction (S.F.;[66]). The results presented in Supplementary Fig. 12, for each modality (transcriptomics and structural connectivity), and across the two metrics (Q_mod and S.F.), establish the extent to which the networks derived from mitochondrial features are more tightly knit than expected by chance ($p < 0.05$).

### Genome-wide differential gene expression analyses on brain networks

Differential gene expression analysis between the three identified networks was performed using the in situ hybridization RNA transcript abundance data from the Allen Mouse Brain Atlas[64,107] accessed via The Harmonizome[108] (https://maayanlab.cloud/Harmonizome/dataset/Allen+Brain+Atlas+Adult+Mouse+Brain+Tissue+Gene+Expression+Profiles). Starting from all microscopic areas with in situ gene expression data ($n = 2170$), areas that correspond to each of our target brain areas were averaged to obtain a single expression value for each gene, per brain area (details in Supplementary Data 1). The atlas data did not differentiate between dorsal and ventral dentate gyrus, so for these analyses the two areas were combined into one. We then collapsed all areas of each of the network 1 ($n = 6$ areas), network 2 (6 areas), and network 3 (4 areas) to create average gene expression values for each network. To compare networks to one another, we used a threshold of one Log2 unit, yielding genes either *over*expressed by >100% (double) or under-expressed by >50% (half) in one network relative to the other two networks. Using the resulting list of genes either over- or under-expressed in each network (Supplementary Data 2), we performed gene ontology (GO) analysis to ask which biological processes are enriched (FDR $p < 0.05$) in each mito-based brain network, and then used a network-based approach to group the resulting biological processes into a few simple categories of pathways in ShinyGo 0.76.3. We report separately the enriched categories for network 1 relative to networks 2 + 3 combined (Fig. 4a), and for networks 2 and 3 relative to other networks (Supplementary Fig. 13).

### Transcriptional mitochondrial phenotypes and specialization

For the analyses of mitochondrial phenotypes (i.e., mitotypes), we mapped 946 out of 1136 mitochondrial genes listed in MitoCarta 3.0[16] to the Allen Mouse Brain Atlas gene expression dataset. Supplementary Data 3 (R Markdown) includes a full list of mitochondrial genes identified and those that could not be mapped. To compare the mitochondrial gene and pathway signatures between 16 main areas (dorsal and ventral DG combined) from networks 1, 2 and 3, we used *Funny-Looking Kid* in R version 4.2.1. We also performed a test of robustness and sensitivity analysis by repeating these analyses using all microscopic sub-areas that were averaged into our main 16 areas, which provided additional evidence for the distinct mitochondrial phenotypes among network 1 sub-areas compared to other areas (Supplementary Data 3). For the pathway-specific analysis, each mitochondrial gene was assigned to a mitochondrial pathway ($n = 149$) using MitoCarta3.0 annotations. The resulting data was z-score transformed with a mean of 100 and a standard deviation of 10 (i.e., a transformed z-score) to allow for direct gene expression comparisons between brain areas. From the transformed data, the expression of genes in a given pathway were averaged, yielding 149 mitochondrial pathway scores for each brain area. Hierarchical clustering (Fig. 4c) of the resulting matrix (16 brain areas x 149 pathways) was performed using the Euclidean distance calculated from relative pathway scores and the *ward.D2* method.

For the differential analysis of mito-pathways across networks, a mean mitochondrial pathway score was calculated by taking the average scores for all areas belonging to network 1 and networks 2 + 3, and log2 fold difference values were used to rank pathways from lowest (lower in network 1) to highest (higher in network 1). Bivariate plots as in reference[32] were used to visualize area-specific transcriptional mitochondrial phenotypes and to quantify the relative magnitude of mitochondrial specialization between brain areas, using two top and two bottom pathways, color-coded by network. A detailed description of the methods, including the code used in the analyses and more detailed interactive plots with sub-areas of the 16 main analyzed areas, is available as an R markdown file (Supplementary Data 3).

### Statistical analyses

Standardized effect sizes (Hedge's g) were computed to quantify the effect of stress conditions on mitochondrial features. Significant effect sizes were determined by the 95% confidence intervals. Two-way

ANOVAs were used to compare the effects on a variable between groups. Frequency distributions of the effect sizes for the two stressors and of the mitochondrial-behavior correlations were fitted with Gaussian curves and analyzed by one-sample $t$-tests compared to the null distribution. A survival curve using Mantel-Cox log-rank test was generated for novelty suppressed feeding test latencies because the test is capped at 600 s. Behavioral scores between groups were analyzed using Tukey's multiple comparison ordinary one-way ANOVA. Correlations between behavioral scores and mitochondrial activities were estimated using Spearman's $r$ ($r$) to guard against inflation. Correlations between tissues' mitochondrial activities were measured as Pearson's $r$, and hierarchical clustering was performed on the data using Euclidian distance with Ward's clustering algorithm. $T$-tests were used to compare groups' correlations. Permutation testing was used to analyze mitochondrial features between vs. within brain areas and to analyze the mitochondrial networks against gene co-expression and structural connectome data. To assess the group (CORT vs. CSDS) differences in the regional response to stressors are statistically robust, a phase randomized (PR) null approach[109] was used, where we generated 1000 iterations of null data separately for CORT and CSDS groups and ran our TDA-based Mapper approach on each null dataset. PR null preserved the covariance across mice but shuffled any relation between regional responses. Non-parametric permutation statistics were later estimated for each group. Statistical analyses were performed with Prism 9 (GraphPad), Matlab R2021b, and Metaboanalyst version 4[110]. Mitochondrial phenotype analyses were performed in R version 4.2.1 (2022-06-23).

### Reporting summary

Further information on research design is available in the Nature Portfolio Reporting Summary linked to this article.

## Data availability

The datasets generated during and analyzed during the study are available in the Source Data file or Supplementary Information files. This study used various publicly available datasets. We used the Anatomic Gene Expression Atlas from the Allen Brain Atlas, which can be accessed at https://mouse.brain-map.org/agea and the Allen Mouse Brain Connectivity Atlas, which can be accessed at https://connectivity.brain-map.org/. Additionally, in situ hybridization RNA transcript abundance data was obtained from the Allen Mouse Brain Atlas accessed via The Harmonizome, (https://maayanlab.cloud/Harmonizome/dataset/Allen+Brain+Atlas+Adult+Mouse+Brain+Tissue+Gene+Expression+Profiles). We also utilized Mitocarta 3.0, which can be accessed at https://www.broadinstitute.org/mitocarta/mitocarta30-inventory-mammalian-mitochondrial-proteins-and-pathways. Source data are provided with this paper.

## Code availability

For TDA Mapper analysis, the generic toolbox is available at https://braindynamicslab.github.io/dyneusr/. The software used for multi-slice network analysis can be accessed at https://github.com/GenLouvain/GenLouvain. Custom scripts generated for this paper can be accessible from Saggar, M. Brain mitochondrial diversity and network organization predict anxiety-like behavior in male mice. *Zenodo*, https://doi.org/10.5281/zenodo.7955250, (2023)[111].

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

## Acknowledgements

Work of the authors was supported by NIH grants R01GM119793, M.P., R01MH122706, M.P., R00MH108719, C.A., P50MH090964, C.A., R01MH126105, C.A., R01MH111918, D.D., DP2MH119735, M.S., and the Baszucki Brain Research Fund, M.P. The authors are grateful to Christoph

Kellendonk for advising on the selection of brain areas and for commenting on this manuscript, David Sulzer and René Hen for providing infrastructure that supported this work, and Amira Millette for technical assistance. Portions of figures were created with BioRender.com.

## Author contributions

M.P., C.A., D.D. conceived the project. A.M.R., P.R., C.A. collected the data from the mouse cohort. A.J. performed validation biochemical experiments. A.M.R., P.R., A.L., M.S. analyzed the molecular and enzyme activity data. A.S.M. and J.D. performed the gene expression-based mitotyping analyses. A.M.R., M.S., A.S.M., J.D., P.R., A.J. generated the figures. A.M.R., M.S., A.S.M., C.S., E.V.M., D.D., C.A., M.P. interpreted the data. A.M.R. and M.P. drafted the manuscript. All authors reviewed and edited the final version of the manuscript.

## Competing interests

The authors declare no competing interests related to this work. C.A. receives research funding from Sunovion Pharmaceuticals.

## Additional information

¹Division of Behavioral Medicine, Department of Psychiatry, Columbia University Irving Medical Center, New York, NY, USA. ²Department of Psychiatry and Behavioral Sciences, Stanford University, Stanford, CA, USA. ³Columbia University Institute for Developmental Sciences, Department of Psychiatry, Columbia University Irving Medical Center, New York, NY, USA. ⁴Department of Biological Sciences, Columbia University, New York, NY, USA. ⁵Division of Systems Neuroscience, Department of Psychiatry, Columbia University Irving Medical Center, New York, NY, USA. ⁶Brain Mind Institute, Ecole Polytechnique Federal de Lausanne (EPFL), Lausanne, Switzerland. ⁷Division of Molecular Therapeutics, Department of Psychiatry, Columbia University Irving Medical Center, New York, NY, USA. ⁸New York State Psychiatric Institute, New York, NY, USA. ⁹Department of Pediatrics, Columbia University Irving Medical Center, New York, NY, USA. ¹⁰Division of Developmental Neuroscience, Department of Psychiatry, Columbia University Irving Medical Center, New York, NY, USA. ¹¹Department of Neurology, H. Houston Merritt Center, Columbia Translational Neuroscience Initiative, Columbia University Irving Medical Center, New York, NY, USA. ¹²Robert N Butler Columbia Aging Center, Columbia University Mailman School of Public Health, New York, NY, USA. ✉e-mail: martin.picard@columbia.edu

