## [Peer Review File · Nature Communications]

Brain mitochondrial diversity and network organization predict anxiety-like behavior in miceREVIEWER COMMENTS

Reviewer #1 (Remarks to the Author):

In this manuscript, Rosenberg and colleagues investigated the effect of two different types of stress on the mitochondrial energetic capacity in different brain areas of the mice. They found significant alterations in some parameters that were associated with some brain areas, leading them to propose the occurrence of specific neural circuits related to the stress response according to their mitochondrial energy fitness. The work is nicely designed, the methodological approaches are robust, and the analysis of the data is rigorous. Thus, the overall message of the study is well supported by the data presented. However, there are a few concerns that, whilst they do not invalidate the main approach and conclusions, they should be taken into account by the authors in order to further increase the scientific accuracy.

Comments

1. As the authors are aware, any biochemical parameter analyzed in a piece of brain tissue will be the reflection of the integrated value from different cell types. For instance, the energetic reliance on OXPHOS is higher in neurons than in astrocytes (Bonvento & Bolaños, 2021, *Astrocyte-neuron metabolic cooperation shapes brain activity*. *Cell Metab.* 33:1546-1564, doi: 10.1016/j.cmet.2021.07.006, PMID: 34348099), a circumstance that is reflected in their strongly different structural organization of the mitochondrial respiratory chain and bioenergetic efficiency (Lopez-Fabuel et al. 2016, *Complex I assembly into supercomplexes determines differential mitochondrial ROS production in neurons and astrocytes*, *Proc Natl Acad Sci U S A.* 113(46):13063-13068, doi: 10.1073/pnas.1613701113, PMID: 27799543). Accordingly, the measure of the RC and/or MHI in a brain tissue containing different proportions of these cell types will not allow distinguishing selective effects of the stressors on neurons and in astrocytes, which might be different or even opposite hence resulting in the absence of measurable changes in the tissue. Alternatively, the occurrence of a particular change observed in MHI may be the result of a major effect of the stressor on one cell type in one brain region, and a major effect of the stressor on another cell type in another brain region, which might give rise to the proposal of different mito-networks in behavior to those herein identified. Admittedly, the design of the study is already very robust and likely at its maximal sensitivity capacity, hence with the same methodological approach, it will be extremely challenging to address this concern. However, the authors should at least comment and discuss this potential problem in the manuscript.

2. The findings shown in Figure 2, described in page 11, are very interesting and important, indeed suggesting that the brain regions might be mapped according to their energetic efficiencies. In addition, the authors discuss in several places in the manuscript the occurrence of a diversity of bioenergetically distinct mitochondria in several brain regions and tissues within an individual. In fact, a previous (and recent) study already demonstrated that different tissues segregate their mitochondria towards a particular, energetically characteristic, mitochondrial type, a process that is driven by the cellular -not tissular- metabolic program (Lechuga-Vieco et al. 2020, *Cell identity and nucleo-mitochondrial genetic context modulate OXPHOS performance and determine somatic heteroplasmy dynamics*. *Sci Adv.* 6:eaba5345, doi: 10.1126/sciadv.aba5345, PMID: 32832682). Moreover, neurons show different preference than astrocytes for actively segregating a particular mitochondrial type according to age (PMID: 32832682). Therefore, the authors should discuss these specific findings in the context of these previous observations (PMID: 32832682) to acknowledge the work by others at identifying tissular and cellular mitochondrial segregation.

3. The paragraph and conclusion "protein levels do not reflect enzymatic activity" would need to be re-considered. Obviously, the RC enzymatic activity reflects energy production capacity better than protein levels, as the authors state (page 7). However, the lack of correlation between SDH enzymatic activity and the protein abundance of a SDH subunit, on its own, does not seem to be sufficient to claim that the RC activity determination is not a reflection of the amount of enzymatic protein. First, the authors only focused on one of the several RC enzymes that they analyzed hence the conclusion might be applicable only to SDH; and secondly, the fact that one SDH subunit does not correlate with SDH activity does not make it extensive to the other SDH subunits -i.e., there might be changes in the protein abundances of the other SDH subunits, which were not

analyzed. Accordingly, to be accurate, the authors should not make this strong assertion, and leave their conclusions relied upon the RC specific enzymatic activity values, which -unless otherwise specifically demonstrated- is a direct reflection of the enzyme abundance and perfectly suitable to maintain most -if not all- of their current conclusions.

4. The conclusions of the manuscript are obtained from mitochondrial health parameter herein defined on the bases of RC specific activities, reflecting maximal energetic capacity, as the authors state. However, factors such as substrate availability are known to induce regulatory changes in the mitochondrial machinery, particularly, the structural assembly of the RC, that, in the absence of alterations of their specific activities, profoundly changes the bioenergetic efficiency of mitochondria (Guaras et al. 2016, *The CoQH2/CoQ Ratio Serves as a Sensor of Respiratory Chain Efficiency*.

Cell Rep. 15:197-209, doi: 10.1016/j.celrep.2016.03.009, PMID: 27052170). This is a critical point that the authors should discuss in the manuscript. Moreover, in relation to this issue, the authors should mention and/or discuss to what extent the administration of corticosterone, known for its strong effects on peripheral glucose uptake hence likely affecting brain glucose availability and therefore mitochondrial respiratory chain complexes assembly and bioenergetic efficiency.

5. The description of the enzymatic activity determinations method should be provided in more detail, and not just cited under Ref. 28, specifically regarding the substrates used. For instance, complex I activity is mentioned as "NADH dehydrogenase", but it should be "NADH-ubiquinone oxidoreductase". Similarly, the description of SDH and complex II should be clearly defined as "succinate-ubiquinone oxidoreductase" and explicitly indicated that both nomenclatures (SDH and complex II) define the same enzymatic activity.

Reviewer #2 (Remarks to the Author):

Rosenberg, et al. present an expansive and rigorous examination of mitochondrial function across brain regions and peripheral tissues under two different behavioral conditions (CSDS and chronic CORT). The authors use summary indices of both mitochondrial function and anxiety-like behaviors collapsing multiple variables to understand the relationship between behavior, mitochondrial function and tissue specificity, concluding that there a functionally meaningful brain networks with covarying mitochondrial activity. This work represents both technical innovations in miniaturizing and increasing throughput for mitochondrial enzymatic activity assays and a comprehensive statistical analysis to integrate data across tissues and behavioral batteries. The growing interest in brain mitochondria makes this both a timely and important manuscript.

Suggestions for improvement:

A main concern with the data as it is presented is that all of the analyses shown depend on the mitochondrial enzymatic assays that are stated as newly developed in their high-throughput miniaturized state for this manuscript, yet data from these assays is only shown in aggregate and correlation matrices. For proof of principle, the authors should show representative data from the miniaturized assays (at least one example brain region with the 4 enzymatic assays) and explain how this mini assay compares to other assays that have been run in frozen brain tissue samples from similar behavioral patterns.

The authors make an interesting point about the importance of measuring enzymatic function over protein or other expression metrics for representing mitochondrial health and show data related to this claim in Extended Data Figure 3. It is unclear how enzymatic activity was measured in this slice assay. Further details should be added to the methods, as well as representative raw data of protein and enzymatic function should be included in addition to the correlations shown. The robustness of the data behind this claim could influence standards in mitochondrial data reporting in the field.

Minor points

It is ambiguous what the pairwise comparisons described in Figure 1B are and where the significance cut off point is on the purple spectrum.

In Figure 2C, please note in the figure legend that the yellow boxed regions correspond to graphed correlation data.

Extended data figure 1A is stated to illustrate the difference between mitochondrial copy number by cell vs by density, but the diagram does not provide clarification on this front, as only number per cell is represented. Expanding the diagram to include a representation of per volume measures would be useful

In extended data figure 1C, please add yellow boxes around correlations shown in D to match the format of main Figure 2

The network analysis shown in Extended Data Figure 6B is not discussed in the text and not sufficiently described in the figure legend with respect to interpretation of the results shown. The interpretation for this analysis should be discussed, or the panel should be removed.

The clarity of Extended data Figure 7 would be greatly improved by consistency of color coding of groups. In panel B CSDS animals are shown in green and recovered in blue, but in panel D CSDS is red and recovered is green. In panel C the CORT group is shown in red, but is beige in panel D. Further, the authors should confirm that in panel C CORT is the correct group to be displayed here, as the text says comparisons are to the naïve mice.

Reviewer #3 (Remarks to the Author):

Brief summary:

The authors consider a 6-dimensional mitochondrial profile for 17 brain regions for a range of mice with and without stressors.

A few comments:

I think that there are lots of interesting results here. For me a number of the most compelling results were in the aggregated datasets (e.g. 1d, 2d) — these results were somehow lost in comparison to the fancier analysis types. I was concerned about risk of over-interpretation for the disaggregated aspects of the analysis. I unpack some of these concerns below. I think subsequent versions of the manuscript should place more emphasis on the aggregate results. There are so many results here that it might be helpful for the authors to rank their results by their belief that they are a) likely to be reproduced and b) likely to inform future science. I think more lowly ranked results can then be moved to the supplement to increase the salience of the key take-homes. It might be that taking a more univariate perspective on the data — trying to leverage the 571 samples appropriately — could yield even more insights than already presented.

I really liked the observation “Protein levels do not reflect enzymatic activity ” — it was at odds with my intuition. If the authors care to expand on why this might be in a supplement that would be great (but optional).

I found fig 2d quite compelling. Perhaps it could be promoted in the manuscript. It’s a simple claim but I suspect a robust one.

Figure 1 e,f,g is a kind of exploratory data analysis. I’m not sure I find this compelling. To me there is no really strong effect that is being exposed here. If a really strong effect is not revealed then one might worry that another, similar, analysis method might yield different results: methodological idiosyncrasies might be creating apparent signal.

It looks like lots of hypotheses are being tested. It’s great that the study is systematic but it’s also

true that (as far as I can make out) there aren't lots of mice in each arm. Could the authors clarify throughout when/if they controlled for multiple hypothesis testing? E.g. is this claim "The amygdala (Amyg) showed the greatest CORT-induced increase in CII enzymatic activity (+49%, $p < 0.05$), whereas the 9 periaqueductal gray (PAG) showed the largest CSDS-induced decrease in CI activity (-42%, $p < 0.05$). " robust to multiple hypothesis testing?

Minor:

Could a table be provided indicating the number of mice in each arm of the study?

"The average variation for individual RC enzymatic activities and mtDNA density across all animals and all 17 brain regions was 36% (coefficient of variation, C.V.) "
I don't understand — this seems like two quantities?

"the variation in mitochondrial health between mice reached up to 2.9-fold between the animals with the lowest and highest activities "
Is this the variation in the MHI? In general perhaps it's better to use MHI instead of the vaguer phrase mitochondrial health.

"This means that for a given brain region, there are large mouse-to-mouse differences in mitochondrial content and RC activity, even among inbred naïve mice not exposed to stressors "
How does this claim follow from the preceding claims given that they appear to be given for all mice in the study (which include those exposed to stressors)

"suggesting opposing effects of these two different stress models on brain mitochondria "
This was surprising. Is there any account for this?

Ref 50 has come out strangely

"When applied to mitochondrial features, separately for CORT and CSDS, the shape graphs revealed differences in regional mitochondrial recalibrations across the two groups (Figure 1e-f). "
Can you clarify inline what the features are (the figure helps).

Figure 1d makes we wonder whether — CORT and CSDS are possibly similar stressors but the CORT was applied too weakly? Perhaps if e.g. the CORT concentration were increased the two would resemble each other? I'm not proposing the authors perform additional experiments -- but this interpretation would suggest putting less emphasis on CORT in the manuscript.

In 1e "#mice per group " is listed — do you mean that or do you mean something like "mouse index"?

In the figure/picture 1e readers will understand what you're trying to do if you call the mapper output something like "mapper output illustration"

In 1e/1f — it might help the reader if you clarify in the maintext/captions what the nodes are — are they mice, features, clusters of mice, clusters of features or none of the above?

A challenge in using the data visualization in 1e,f is that if the reader lacks clear expectations as to what a null model would produce — or is unfamiliar with this particular TDA tool — then they can prove inscrutable. I'd find the conclusions in 1g more convincing if the data were shuffled in an appropriate manner and comparisons to this null made.

The authors use Pearson's r — why not use Spearman's rank?

"Previous research in rodents showed that mitochondrial function in specific brain regions, such as the NAc, is linked to complex behaviors such as social dominance and anxiety^{4,29}. Here we confirm the strong correlation between NAc mitochondrial health and anxious behavior, as the strongest correlation between EPM-based anxiety-like behavior and MHI was in the NAc. "
To me this seems an important corroborative result perhaps this can be emphasised more earlier in the text?

“the data matrix for each group was process through the TDA-based Mapper pipeline ”

Reviewer #4 (Remarks to the Author):

Mitochondria are essential to proper nervous system functioning with increasing evidence of their functional diversity across distinct cell types, physiologic coupling to neuronal activity, and deficits in neuropsychiatric disease. In this study, Rosenberg, Sagar et al. sought to determine associations between brain region-specific mitochondrial function and animal behavior.

The authors obtained biochemical measurements of mitochondrial respiratory chain function and proxies for mitochondrial content from 17 brain regions of 27 adult male mice that underwent standard behavioral phenotyping. To increase behavioral diversity, 17 of these mice were exposed to models of chronic stress prior to behavioral testing. They identified some significant correlations between mitochondrial functioning in certain brain regions with behavioral metrics (e.g. their derived mitochondrial health index score in nucleus accumbens is positively correlated with anxiety-like behavior across animals ascertained by the elevated plus maze assay). Network analyses on this mitochondrial functional data further showed clusters of brain regions that mirror known neuronal functional networks (e.g. cortico-striatal) and appeared to overlap with mouse structural connectivity and gene coexpression networks.

Overall, the authors have generated a novel dataset providing intriguing associations between mitochondrial function and neurobiological output (i.e., animal behavior). However, it is difficult to interpret the biological meaning of these associations and their ultimate validity given limited follow-up experiments and validation of the original dataset.

Suggestions for improvements:

1) A more thorough investigation on the possible biological underpinning of the variance in mitochondrial functioning observed across brain regions would strengthen the study. As the authors mention, there is large cellular compositional differences across the 17 analyzed brain regions with distinct neuronal types and varying proportions of glial cells. As there appears to be brain cell-type specific mitochondrial functional differences (e.g. Fecher et al. Nat Neuro 2019), could part of the differences observed here be based on variation in neuronal/glial composition? To investigate the contribution of cellular composition, one could use existing single-cell RNA-seq atlases from mouse to deconvolute the Allen Brain mouse gene expression atlas used here to get estimates of the major cell type abundances in these 17 brain regions and look for associations to mitochondrial functioning. Similarly, they could use the Allen Brain mouse atlas to look for correlations with regional gene expression to nominate putative biological pathways implicated in the mito-networks (e.g. could further investigate the neuroendocrine hypothesis stated in the discussion). The authors also suggest metabolic coupling with neuronal activity as another possible mechanism. Although technically challenging, it would be interesting to directly test this idea using neuronal activation/inhibition strategies.

2) Experimental validation of select findings would strengthen the evidence of the data collected in the initial dataset. For example, orthogonal experiments to confirm associations of amygdala/PAG mitochondrial function and chronic stress models; NAc mitochondria function and anxiety-like behavior; SN mitochondria function and social interaction. The presented results would also be more convincing if an identified association was shown to be causally linked (as done in prior work, Hollis PNAS 2015).

3) If I understand the presentation of the results correctly, the vast majority of the associations depicted in the heatmaps in Fig. 1a (side note: are these values corrected for multiple testing?) and Fig. 2c are non-significant. If this is the case, I find the descriptions of the data in the results section a bit odd. E.g. “CORT-treated mice had higher mitochondrial activities than non-stressed animals in ~60% of brain regions, whereas CSDS animals had lower activities in ~82% of brain regions”; but based on Fig.1a only 1 region with 1 metric (out of possible 102 comparisons) was significantly increased in CORT-treated mice. Another example: “For both OFT and EPM, higher mitochondrial health in the majority of brain regions was correlated with higher avoidance scores

(average $r=0.12, 0.34$ respectively)”; but only 3 associations in Fig 2c were significantly correlated. In general, how are we supposed to interpret non-significant associations?

4) Related to point (3), perhaps a clearer presentation of the statistical methods used would help clarify the findings. E.g. highlighting statistically significant associations (and how that’s defined) for each of the heatmaps.

5) Although not my area of expertise, it was not clear to me why mtDNA density would be a ‘more appropriate and generalizable estimate of mitochondrial genome abundance across mouse brain regions’ compared to mtDNA_{cn}. Wondering about how choice of this metric might introduce more/less confounding with cellular compositional differences across brain regions (naively, I would assume mtDNA_{cn} is better corrected for potential compositional differences).

6) It was also unclear why CSDS model would create such a coherent and opposite effect on mitochondrial functioning compared to CORT-treated mice. Furthermore, from the available data presented in Ext. Data Fig. 8 it appears these animals performed similarly on behavioral tests. Not sure how to reconcile the potential disparate effects on mitochondria functioning seen in these models yet similar behavioral outcomes.

Please note that we have included a new figure as Extended Data figure 1, so all previous Extended Data Figure numbers have been shifted by one. The figure numbers in the reviewer comments refer to the previous figure numbers, whereas the numbers in the responses refer to the new figure numbers.

Reviewer #1 (Remarks to the Author):

In this manuscript, Rosenberg and colleagues investigated the effect of two different types of stress on the mitochondrial energetic capacity in different brain areas of the mice. They found significant alterations in some parameters that were associated with some brain areas, leading them to propose the occurrence of specific neural circuits related to the stress response according to their mitochondrial energy fitness. The work is nicely designed, the methodological approaches are robust, and the analysis of the data is rigorous. Thus, the overall message of the study is well supported by the data presented. However, there are a few concerns that, whilst they do not invalidate the main approach and conclusions, they should be taken into account by the authors in order to further increase the scientific accuracy.

Comments

1. As the authors are aware, any biochemical parameter analyzed in a piece of brain tissue will be the reflection of the integrated value from different cell types. For instance, the energetic reliance on OXPHOS is higher in neurons than in astrocytes (Bonvento & Bolaños, 2021, Astrocyte-neuron metabolic cooperation shapes brain activity. *Cell Metab.* 33:1546-1564, doi: 10.1016/j.cmet.2021.07.006, PMID: 34348099), a circumstance that is reflected in their strongly different structural organization of the mitochondrial respiratory chain and bioenergetic efficiency (Lopez-Fabuel et al. 2016, Complex I assembly into supercomplexes determines differential mitochondrial ROS production in neurons and astrocytes, *Proc Natl Acad Sci U S A.* 113(46):13063-13068, doi: 10.1073/pnas.1613701113, PMID: 27799543). Accordingly, the measure of the RC and/or MHI in a brain tissue containing different proportions of these cell types will not allow distinguishing selective effects of the stressors on neurons and in astrocytes, which might be different or even opposite hence resulting in the absence of measurable changes in the tissue. Alternatively, the occurrence of a particular change observed in MHI may be the result of a major effect of the stressor on one cell type in one brain region, and a major effect of the stressor on another cell type in another brain region, which might give rise to the proposal of different mito-networks in behavior to those herein identified. Admittedly, the design of the study is already very robust and likely at its maximal sensitivity capacity, hence with the same methodological approach, it will be extremely challenging to address this concern. However, the authors should at least comment and discuss this potential problem in the manuscript.

Response: This is an excellent point. While our brain biochemistry approach cannot directly address the point around cell types, in order to address the reviewer's point we have performed a new analysis which involves correlating cell type abundances with mitochondrial features. We have obtained data for the various cell type abundances across each brain region from the Blue Brain Cell Atlas (Ero et al., 2018; Keller et al., 2018), including total cell density, neuronal density, and neuroglial densities. Specifically, we also included densities of excitatory, inhibitory, and modulatory cells in addition to densities of oligodendrocytes, astrocytes, and microglia. We correlated the density measures for each cell type from each brain region with the average (pooled average across all animals) of each mitochondrial measure for each region (Figure below). The left-hand side of the correlation matrix shows the associations in cell type abundances between brain regions. The right-hand side (indicated by the yellow box) shows the strongest associations between cell type abundances and mitochondrial measures.

The strongest correlations are between astrocyte density and citrate synthase activity ($r^2 = 0.58$, $n=16$ regions, $p<0.001$, graph A, *right*) and oligodendrocyte density (Oligos) and complex II activity ($r=0.48$, $n=16$, $p=0.003$, graph B, *right*). However, there were no association between the abundance of any cell type and MHI, our composite marker of RC capacity (last column on the right of the heatmap).

The sample size for these comparisons ($n=16$ regions) is fairly limited, particularly in light of the multiple correlations. The correlative nature of this data also does not provide definitive data on the mitochondrial properties of brain cell types. Finally, it should be noted that the design of our assays, particularly the partitioning of samples into batches (1 batch = 96-well plate with a maximum of 32 samples, in triplicates), were optimized to detect differences between treatment groups, and to avoid the potential confound of batches in the analyses of mouse-to-mouse differences in mitochondrial metrics, with behavior. As a result, we put less confidence in the direct comparisons between brain regions in this manuscript, and consequently have avoided reporting these analyses in the manuscript, instead focusing on outcomes for which the study design is optimal.

Nevertheless, should the reviewer find these data relevant and deemed them worthy of inclusion in the manuscript, we would be happy to fulfill this request and include it.

Thus, to explicitly address the lack of cell type specificity noted by the reviewer and other limitations of this work, we have added a limitations section at the end of the discussion, which reads as follows (p. 21):

Limitations. Notable limitations of this study include the lack of cell type specificity. Neurons operate in a metabolic partnership with astrocytes and glial cells (Bonvento and Bolanos, 2021), and different cell types exhibit different molecular mitochondrial phenotypes (e.g., (Lopez-Fabuel et al., 2016)) that cannot be disentangled in tissue homogenates. While neither enzymatic or functional mitochondrial profiling at the single brain cell is currently technically feasible, it remains possible that mitochondrial phenotypes between brain regions are influenced by differences in cell type proportions. Moreover, our molecular and biochemical mitochondrial phenotypes do not reflect other factors that

can influence the efficiency or activity of the mitochondrial RC or OxPhos system *in vivo*, such as variations in cofactor abundance and RC structural assembly (e.g., supercomplexes)(Guaras et al., 2016). Because functional assays require harvesting brain tissue, it also was not feasible to ascertain within a given animal how stable (trait) or dynamic (state) the brain biochemical mitochondrial phenotypes are. If mitochondrial phenotypes were more dynamic than expected, our estimated proportion of behavioral variance attributable to mitochondrial biology could be substantially underestimated.

2. The findings shown in Figure 2, described in page 11, are very interesting and important, indeed suggesting that the brain regions might be mapped according to their energetic efficiencies. In addition, the authors discuss in several places in the manuscript the occurrence of a diversity of bioenergetically distinct mitochondria in several brain regions and tissues within an individual. In fact, a previous (and recent) study already demonstrated that different tissues segregate their mitochondria towards a particular, energetically characteristic, mitochondrial type, a process that is driven by the cellular -not tissular- metabolic program (Lechuga-Vieco et al. 2020, Cell identity and nucleo-mitochondrial genetic context modulate OXPHOS performance and determine somatic heteroplasmy dynamics. Sci Adv. 6:eaba5345, doi: 10.1126/sciadv.aba5345, PMID: 32832682). Moreover, neurons show different preference than astrocytes for actively segregating a particular mitochondrial type according to age (PMID: 32832682). Therefore, the authors should discuss these specific findings in the context of these previous observations (PMID: 32832682) to acknowledge the work by others at identifying tissular and cellular mitochondrial segregation.

Response: We have corrected this omission and highlight the significance of this finding in relation to the work by Lechuga-Vieco, Enriquez and colleagues (p.12): “This finding aligns with recent work in mice showing that genetically-driven mitochondrial phenotypes exhibit strong segregation between different cell types and tissues⁵⁴, and has two implications. [...]”

3. The paragraph and conclusion “protein levels do not reflect enzymatic activity” would need to be re-considered. Obviously, the RC enzymatic activity reflects energy production capacity better than protein levels, as the authors state (page 7). However, the lack of correlation between SDH enzymatic activity and the protein abundance of a SDH subunit, on its own, does not seem to be sufficient to claim that the RC activity determination is not a reflection of the amount of enzymatic protein. First, the authors only focused on one of the several RC enzymes that they analyzed hence the conclusion might be applicable only to SDH; and secondly, the fact that one SDH subunit does not correlate with SDH activity does not make it extensive to the other SDH subunits -i.e., there might be changes in the protein abundances of the other SDH subunits, which were not analyzed. Accordingly, to be accurate, the authors should not make this strong assertion, and leave their conclusions relied upon the RC specific enzymatic activity values, which -unless otherwise specifically demonstrated- is a direct reflection of the enzyme abundance and perfectly suitable to maintain most -if not all- of their current conclusions.

Response: Thank you for this recommendation. We have reconsidered and significantly toned down the subtitle of this section, and the interpretation of these results (p.7):

*“Protein levels **and** enzymatic activity*

We initially explored if RC protein abundance is a viable surrogate for mitochondrial RC activity (Hollis et al., 2015), which could theoretically allow high spatial resolution imaging of the entire brain. We focused on the cerebellum due to its well-defined cellular composition and laminar organization, where the Purkinje cell layer is flanked by molecular and granular layers (Fecher et al., 2019) (**Extended Data Fig. 4a**). Compared to protein **abundance**, **enzymatic activity** ultimately determines mitochondrial RC function and energy production capacity and consequently should be regarded as the most representative measure of mitochondrial **phenotypes**. In consecutive cerebellar slices, we compared **RC complex II enzymatic activity measured spectrophotometrically**, to the **protein abundance** of a complex II subunit, SDHA (succinate dehydrogenase, subunit A), for which a validated high-affinity antibody **allows its quantification by microscopy** (**Extended Data Fig. 4b-c**). **Across the three cerebellar layers**, enzyme activity did not correlate with protein abundance assessed by immunohistochemistry and densitometry (proportion of shared variance, $r^2=0.02-0.07$), indicating that protein abundance **and enzymatic activity are not equivalent** (**Extended Data Fig. 4d**). The reasons for this finding could include the action of post-translational modifications, stoichiometry of the four SDH subunits, or the biochemical context that drive **biochemical** activity independent of protein content (e.g., (Stepanova et al., 2016)). Therefore, we focus **all** downstream analyses on direct measures of mitochondrial RC enzymatic activity and mtDNA density.”

4. The conclusions of the manuscript are obtained from mitochondrial health parameter herein defined on the bases of RC specific activities, reflecting maximal energetic capacity, as the authors state. However, factors such as substrate availability are known to induce regulatory changes in the mitochondrial machinery, particularly, the structural assembly of the RC, that, in the absence of alterations of their specific activities, profoundly changes the bioenergetic efficiency of mitochondria (Guaras et al. 2016, The CoQH2/CoQ Ratio Serves as a Sensor of Respiratory Chain Efficiency. Cell Rep. 15:197-209, doi: 10.1016/j.celrep.2016.03.009, PMID: 27052170). This is a critical point that the authors should discuss in the manuscript. Moreover, in relation to this issue, the authors should mention and/or discuss to what extent the administration of corticosterone, known for its strong effects on peripheral glucose uptake hence likely affecting brain glucose availability and therefore mitochondrial respiratory chain complexes assembly and bioenergetic efficiency.

Response: Thank you for bringing this paper to our attention. We have included it in the limitations section (as included in response to point 1) and broader discussion of the factors that may influence mitochondrial OXPHOS capacity.

We now also mention the metabolic consequence of corticosterone on p.16:

“This difference in mitochondrial recalibrations between both stress models may be driven by several factors. This includes the stressor duration, although the temporal dynamics over which stress-induced mitochondrial recalibrations take place remain poorly defined and will require further focused attention. Differences in the effects of CORT vs CSDS on mitochondria also could be related to their peripheral effects or neuroendocrine underpinnings (single hormone for CORT vs multiple physiological neuroendocrine recalibrations for CSDS), regional differences in glucocorticoid and mineralocorticoid receptor density, or other factors generally relevant to chronic stress rodent models.”

5. The description of the enzymatic activity determinations method should be provided in more detail, and not just cited under Ref. 28, specifically regarding the substrates used. For instance, complex I activity is mentioned as “NADH dehydrogenase”, but it should be “NADH-ubiquinone oxidoreductase”. Similarly, the description of SDH and complex II should be clearly defined as “succinate-ubiquinone oxidoreductase” and explicitly indicated that both nomenclatures (SDH and complex II) define the same enzymatic activity.

Response: We have corrected the nomenclature for complex I (NADH-ubiquinone oxidoreductase) and complex II (succinate-ubiquinone oxidoreductase). We have insured that the methods include all details including the substrates used (p.28-30):

Citrate synthase (CS) enzymatic activity was determined by measuring the increase in absorbance of DTNB at 412nm at -30°C in 200µL of a reaction buffer (200 mM Tris, pH 7.4) containing acetyl-CoA 10 mM, 10 mM 5,5'- dithiobis- (2-nitrobenzoic acid) (DTNB), 2 mM oxaloacetic acid, and 10% w/v Triton-x-100. The rate of conversion of DTNB into NTB²⁻ ions indicates the enzymatic activity and is used as a marker of mitochondrial content. Oxaloacetate is removed from the assay mix as a way to measure non-specific activity. The final CS activity was determined by integrating OD412 change over 150-400 seconds and then subtracting the non-specific activity. 10µL of homogenate was used to measure CS activity.

Complex I (CI, NADH-ubiquinone oxidoreductase) activity was determined by measuring the decrease in absorbance of DCIP. The rate of absorbance of DCIP is measured at 600nm at 30°C, in 200µL of a reaction buffer (potassium phosphate 100mM, pH 7.4) containing 550mg/ml bovine serum albumin (BSA), 50mM potassium cyanide (KCN), 20 mM decylubiquinone, and 0.4mM antimycin A. 10µL of homogenate was used to measure CI activity. Antimycin A and KCN are used to inhibit electron flow through complexes III and IV. The negative control condition includes rotenone (200mM) and piericidin A (0.2mM), which selectively inhibit NADH-ubiquinone oxidoreductase. The final CI activity was determined by integrating OD600 change over 150-500 seconds, and by subtracting the rate of NADH oxidation in the presence of rotenone and piericidin A from the total decrease in absorbance.

Complex II (CII, succinate-ubiquinone oxidoreductase, also known as SDH, succinate dehydrogenase) activity was determined by measuring the decrease in absorbance of DCIP. The rate of absorbance of DCIP was measured at 600nm at 30°C, in 200µL of a reaction buffer (potassium phosphate 100mM, pH 7.4) containing 50mg/mL bovine serum albumin (BSA), 500µM rotenone, 500mM succinate-tris, 50mM potassium cyanide (KCN), 20 mM decylubiquinone, 20mM DCIP, 50mM ATP, 0.4mM antimycin A. 15µL of homogenate was used to measure CII activity. The negative control condition includes sodium-malonate, which inhibits succinate-ubiquinone oxidoreductase. The final CII activity was determined integrating OD600 change over 300-800 seconds, and by subtracting the absorbance in the presence of malonate (500mM) from the total decrease in absorbance.

Complex IV (CIV, also cytochrome c oxidase) activity was determined by measuring the decrease in absorbance of cytochrome c. The rate of conversion of cytochrome c from a reduced to oxidized state was measured at 550nm at 30°C, in 200µL of reaction buffer (100mM potassium phosphate, pH 7.5) containing 10% w/v n-dodecylmaltoside and 120µM of purified reduced cytochrome c. 6µL of homogenate was used to measure COX activity. The negative control condition omits tissue homogenate determine the auto-oxidation of reduced cytochrome c. The final CIV activity was determined by integrating OD550 change over 150-500 seconds, and by subtracting the non-specific activity from the total decrease in absorbance.

Reviewer #2 (Remarks to the Author):

Rosenberg, et al. present an expansive and rigorous examination of mitochondrial function across brain regions and peripheral tissues under two different behavioral conditions (CSDS and chronic CORT). The authors use summary indices of both mitochondrial function and anxiety-like behaviors collapsing multiple variables to understand the relationship between behavior, mitochondrial function and tissue specificity, concluding that there are functionally meaningful brain networks with covarying mitochondrial activity. This work represents both technical innovations in miniaturizing and increasing throughput for mitochondrial enzymatic activity assays and a comprehensive statistical analysis to integrate data across tissues and behavioral batteries. The growing interest in brain mitochondria makes this both a timely and important manuscript.

Suggestions for improvement:

A main concern with the data as it is presented is that all of the analyses shown depend on the mitochondrial enzymatic assays that are stated as newly developed in their high-throughput miniaturized state for this manuscript, yet data from these assays is only shown in aggregate and correlation matrices. For proof of principle, the authors should show representative data from the miniaturized assays (at least one example brain region with the 4 enzymatic assays) and explain how this mini assay compares to other assays that have been run in frozen brain tissue samples from similar behavioral patterns.

Response: Apologies if we mis-represented the assay. The assay used here uses the same chemistry as the original assay. The main difference is that the methodology in this paper uses a 0.2ml reaction instead of the 1ml reaction used for several decades. The smaller volume allows the reaction to be measured in 96-well format, which increases throughput by at least an order of magnitude. We have revised the text to avoid confusion, indicating that this method is being applied to brain tissue for the first time (rather than being an entirely new assay). In relation to validation, we have used this 0.2ml assay in white blood cells (Picard et al., 2018; Rausser et al., 2021), tumor tissues (Bindra et al., 2021), and other mouse/rat tissues. We have previously confirmed that each mitochondrial enzymatic assay scales linearly with increasing tissue sample added to the reaction, and that each assay performs as expected notably with the proportions of specific and non-specific enzymatic activities recorded in various tissues.

To validate the miniaturized assays, we performed each enzymatic assay in both the miniaturized 96-well plate and in the traditional 1ml cuvette assay, using increasing tissue amounts (4, 8, 12, 16, 20ul) from the same brain region (cerebellum) of one mouse (wild type, male). The tissue sample was homogenized according to the method described in the paper, but was diluted 1:200 (weight:volume, mg:μL) to ensure adequate volume. Enzyme activities were quantified spectrophotometrically for CS, CI, SDH, and COX, and were expressed per mg of tissue, as described in the paper. Regressing the increasing tissue amounts with observed enzymatic activities confirmed that a change in tissue amount produced a proportional change in total enzymatic activity ($r^2 = 0.81 - 0.96$; p-value = 0.0032 - 0.037) (Figure below). Further, for all four assays, the 96-well plate generally exhibited less variation between consecutive dilutions compared to the traditional 1ml cuvettes (CS, 8% in the 96-well plate vs. 20% in the cuvette; CI, 15% vs 13%; SDH, 4% vs 9%; COX, 5% vs 9%).

We now include these results and validation in Extended data Fig. 1 (Figure below) and have added a section in the methods (p.28) to describe the methodological details for this validation.

The authors make an interesting point about the importance of measuring enzymatic function over protein or other expression metrics for representing mitochondrial health and show data related to this claim in Extended Data Figure 3. It is unclear how enzymatic activity was measured in this slice assay. Further details should be added to the methods, as well as representative raw data of protein and enzymatic function should be included in addition to the correlations shown. The robustness of the data behind this claim could influence standards in mitochondrial data reporting in the field.

Response: In response to Reviewer 1’s comments, we have substantially revised this section to tone down the claim. In this revision, we have included an example raw spectrophotometric trace for CII activity (Figure below) as Extended Data Fig. 4c.

Sample calculation:

Measure the decrease in absorbance of DCIP (Vmax) during the selected integration time of duplicate samples. Subtract the slope when measured with a complex II inhibitor, to determine specific activity.

The protein abundance was measured by immunofluorescence confocal microscopy in a 200µm cryosection of each mouse brain. In a contiguous 200µm-thick cryosection, tissue punches were collected to perform the spectrophotometric measurements of CII activity, in each of the same animals (as visualized in the diagram below). This analysis brings together the immunofluorescence densitometry (protein abundance) and biochemistry (enzymatic activity) measured through two different modalities. We have clarified this in the results and methods sections and in the figure below, which is now included in Extended Data Fig 4b.

Minor points

It is ambiguous what the pairwise comparisons described in Figure 1B are and where the significance cut off point is on the purple spectrum.

Response: Modified as requested. Only significant (unadjusted p values < 0.05) are on the purple spectrum, and non-significant ones are white/uncolored. The figure legend was modified accordingly. The purpose of the Pairwise comparison is to identify brain regions that were most different from the other regions in their responses to the stressors. This matrix therefore compares the Hedge's g values obtained in Figure 1a between each pair of brain regions. The regions with significant p-values are most different from one another in their response to the stressors.

In Figure 2C, please note in the figure legend that the yellow boxed regions correspond to graphed correlation data.

Response: Modified as requested.

Extended data figure 1A is stated to illustrate the difference between mitochondrial copy number by cell vs by density, but the diagram does not provide clarification on this front, as only number per cell is represented. Expanding the diagram to include a representation of per volume measures would be useful

Response: We have revised Extended Data figure 2a to illustrate the analysis per unit of brain volume. In all panels (b, c, d) the mtDNA_{cn} (per cell) is denoted by a mtDNA with a chromosome, and the mtDNA density is denoted by a mtDNA plasmid alone. Revised panel 2b included below as an example.

In extended data figure 1C, please add yellow boxes around correlations shown in D to match the format of main Figure 2

Response: Added as requested.

The network analysis shown in Extended Data Figure 6B is not discussed in the text and not sufficiently described in the figure legend with respect to interpretation of the results shown. The interpretation for this analysis should be discussed, or the panel should be removed.

Response: We agree with the reviewer. The TDA-based Mapper graph in Extended Data Figure 7b conveys similar information as the graph in Extended Data Figure 8d. We have thus removed panel b from Extended Data Figure 7.

The clarity of Extended data Figure 7 would be greatly improved by consistency of color coding of groups. In panel B CSDS animals are shown in green and recovered in blue, but in panel D CSDS is red and recovered is green. In panel C the CORT group is shown in red, but is beige in panel D. Further, the authors should confirm that in panel C CORT is the correct group to be displayed here, as the text says comparisons are to the naïve mice.

Response: We apologize for the lack of color consistency. We have now made the colors consistent across panels in Extended Data Figure 8b-d.

Reviewer #3 (Remarks to the Author):

A few comments:

I think that there are lots of interesting results here. For me a number of the most compelling results were in the aggregated datasets (e.g. 1d, 2d) — these results were somehow lost in comparison to the fancier analysis types. I was concerned about risk of over-interpretation for the disaggregated aspects of the analysis. I unpack some of these concerns below. I think subsequent versions of the manuscript should place more emphasis on the aggregate results. There are so many results here that it might be helpful for the authors to rank their results by their belief that they are a) likely to be reproduced and b) likely to inform future science. I think more lowly ranked results can then be moved to the supplement to increase the salience of the key take-homes. It might be that taking a more univariate perspective on the data — trying to leverage the 571 samples appropriately — could yield even more insights than already presented.

Response: This point is well taken. We have revised several parts of the manuscript to better emphasize the results that are likely to be reproduced and that will inform future work on brain mitochondrial phenotyping (Figure 1 and 3) in relation to behavior (Figure 2).

I really liked the observation “Protein levels do not reflect enzymatic activity” — it was at odds with my intuition. If the authors care to expand on why this might be in a supplement that would be great (but optional).

Response: Thank you for your interest in this. Reviewer #1 raised good questions about the emphasis that should be put on this portion of the manuscript, in response to which we have toned down our interpretation of these results and the conclusions that we draw from it. Given the complexity of the manuscript, we refrained from expanding on this point.

I found fig 2d quite compelling. Perhaps it could be promoted in the manuscript. It’s a simple claim but I suspect a robust one.

Response: We agree with the reviewer that that the findings displayed in figure 2d are important and compelling. We have therefore expanded on these findings in the results section of the manuscript (p.12):

“As expected, the average correlation between **RC activities** and behaviors was significantly more consistent for brain mitochondria than for mitochondria in peripheral tissues (**Figure 2d**). **For example, whereas mitochondrial phenotypes in several brain regions correlate with anxiety-related behavior on the EPM, mitochondrial measures in the muscles, heart, liver, or adrenal glands of the same animals on average, did not correlate with behavior.** This finding aligns with recent work in mice showing that mitochondrial phenotypes exhibit strong segregation between different cell types and tissues (Lechuga-Vieco et al., 2020), and has two implications. First, it reinforces the specificity of these mito-behavior findings for the brain. Second, it implies that mitochondria across the brain and other tissues within an individual mouse are not equivalent, and likely differentially regulated (Lechuga-Vieco et al., 2020). This naturally raised the question whether specific brain regions within an animal could also exhibit independently regulated mitochondrial properties, and whether brain regions could be functionally organized into separate networks based on their mitochondrial properties.”

Figure 1 e,f,g is a kind of exploratory data analysis. I'm not sure I find this compelling. To me there is no really strong effect that is being exposed here. If a really strong effect is not revealed then one might worry that another, similar, analysis method might yield different results: methodological idiosyncrasies might be creating apparent signal.

Response: We have verified the statistical robustness of this result by running the appropriate graph null models comparing group differences in the regional response to stressors (Sarzynska et al., 2015). Using the phase randomized (PR) null approach (Theiler et al. 1992), we generated 1000 iterations of null data separately for CORT and CSDS groups and ran our TDA-based Mapper approach on each null dataset. PR null preserved the covariance across mice but shuffled any relation between regional responses. Non-parametric permutation statistics were estimated for each group, and we observed the CORT group has a significantly lower participation coefficient (PC) as compared to the null models ($p=0.037$, shown below). Lower PC values suggest higher segregation or (region-specific) influence. A similar analysis for the CSDS group revealed a trend for higher PC as compared to the null models ($p=0.069$). These additional results suggest that our evidence regarding the region-specific stress-induced mitochondrial responses in CORT-treated mice is relatively robust, despite the low number of animals relative to the number of brain regions and mitochondrial measures tested.

While we agree with the reviewer that univariate approaches would provide a simpler picture, unchallenged by multiple testing, our approach is motivated by the growing appreciation of the distributed nature of many brain processes (Cole et al., 2014; Pope et al., 2021; Sporns and Betzel, 2016). Therefore, our view is that this kind of statistically-verified multivariate analysis is conceptually useful, contributing to provide a functionally- and biologically-grounded description of stress-induced brain mitochondrial recalibrations. But in light of this additional information, should the reviewer feel strongly that these data should only be presented as a supplement, we would be happy to more explicitly present these analyses as exploratory.

It looks like lots of hypotheses are being tested. It's great that the study is systematic but it's also true that (as far as I can make out) there aren't lots of mice in each arm. Could the authors clarify

throughout when/if they controlled for multiple hypothesis testing? E.g. is this claim “The amygdala (Amyg) showed the greatest CORT-induced increase in CII enzymatic activity (+49%, $p < 0.05$), whereas the periaqueductal gray (PAG) showed the largest CSDS-induced decrease in CI activity (-42%, $p < 0.05$). ” robust to multiple hypothesis testing?

Response: The reviewer is correct. These are unadjusted p-values and the number of mice per group was relatively limited to make the study feasible (22 tissues per animal, ~570 total samples biochemically assayed). We report the % differences in Figure 1, and the standardized effect sizes (Hedged g, which includes a correction for small samples sizes) in Extended Data Fig. 6, which can be interpreted independent of the p values. If we adjust for multiple testing using the Bonferroni correction across all tissues/brain regions and all mitochondrial measures (22 x 6 = 132), the p value is 0.0038 for an alpha level of 0.05. Given the natural variation between animals, meeting this adjusted significance threshold with the current multi-mitochondrial measures and multi-brain regions design would require hundreds of animals – which is not ethically or technically feasible.

On the other hand, the mitochondrial correlations with behavioral tests in Figure 2 are adjusted for multiple testing. The design here is a little different since we are examining the entire animal population (naïve and stress-exposed animals) and where inter-animal variation provides power to detect true associations between mitochondrial measures and behavior.

Minor:

Could a table be provided indicating the number of mice in each arm of the study?

Response: Thank you for the suggestion. This table is now included as Supplemental Table 1. There were a total of 28 animals (11 naïve and 17 stress-exposed).

Experimental group	CSDS	CSDS-recovered	CORT	Naïve (CSDS-matched)	Naïve (CSDS Recovered)	Naïve (CORT)
Number of animals	6	6	5	3	3	5
Day at sacrifice	14	71	63	14	71	63

“The average variation for individual RC enzymatic activities and mtDNA density across all animals and all 17 brain regions was 36% (coefficient of variation, C.V.) ”

I don’t understand — this seems like two quantities?

Response: The C.V. presented here is an average of all C.V.s for the four RC enzymatic activities and also mtDNA density. We have changed the phrasing of the sentence for clarity (p.8): “The average variation for **all measures (4 individual RC enzymatic activities and mtDNA density)** across all animals and all 17 brain regions was **a C.V. of 36%** (coefficient of variation = **standard deviation / mean**).

“the variation in mitochondrial health between mice reached up to 2.9-fold between the animals with

the lowest and highest activities” Is this the variation in the MHI? In general perhaps it’s better to use MHI instead of the vaguer phrase mitochondrial health.

Response: Revised as requested. We now consistently use MHI.

“This means that for a given brain region, there are large mouse-to-mouse differences in mitochondrial content and RC activity, even among inbred naïve mice not exposed to stressors” How does this claim follow from the preceding claims given that they appear to be given for all mice in the study (which include those exposed to stressors)

Response: The mouse-to-mouse differences (C.V.) for all measures across 17 brain regions among naïve mice alone is 32%, and modestly increases to 36% when including animals exposed to stressors.

“suggesting opposing effects of these two different stress models on brain mitochondria ” This was surprising. Is there any account for this?”

Response: We now note in the manuscript that this may be a difference in the nature of the stressors or may be driven by differences in the duration of the stressors (10 days for social defeat, 63 days for CORT). We further elaborate on this point in our response to question 4 from Reviewer 1.

Ref 50 has come out strangely

Response: This has been corrected.

“When applied to mitochondrial features, separately for CORT and CSDS, the shape graphs revealed differences in regional mitochondrial recalibrations across the two groups (Figure 1e-f). ” Can you clarify inline what the features are (the figure helps).

Response: We have revised text to state that the features here correspond to the five mitochondrial measures plus the mitochondrial health index (MHI) (p.9): “When applied to **the six** mitochondrial features (**i.e., four enzyme activities, mtDNA density, and the MHI**) separately for CORT and CSDS (**both measured as delta of stress vs the naïve group average**), the shape graphs revealed differences in regional mitochondrial recalibrations across the two groups (Figure 1e-f).”

Figure 1d makes we wonder whether — CORT and CSDS are possibly similar stressors but the CORT was applied too weakly? Perhaps if e.g. the CORT concentration were increased the two would resemble each other? I’m not proposing the authors perform additional experiments -- but this interpretation would suggest putting less emphasis on CORT in the manuscript.

Response: There are many factors upon which the CORT and CSDS stressors that were employed here differ from one another, making it difficult to disentangle which factors are the most significant. The nature of the stressors, the duration, and even more minor details such as the housing conditions of the mice are different between the two stressors. While the reviewer makes an interesting point, we believe

that CORT administration duration of almost 8 weeks is rather intense. In most existing studies CORT is administered for 2-6 weeks (Gourley and Taylor, 2009), and 6 weeks of administration has been found to impact behavior and mitochondrial functioning (Xie et al., 2020). We therefore remain cautious in comparing mitochondrial recalibrations in response to the two stressors, but believe that the observed differences are important novel observations that can inform future work in the field.

In 1e “#mice per group ” is listed — do you mean that or do you mean something like “mouse index”? In the figure/picture 1e readers will understand what you’re trying to do if you call the mapper output something like “mapper output illustration”

Response: Thanks for pointing these out. We have modified the figure accordingly to clarify both points.

In 1e/1f — it might help the reader if you clarify in the main text/captions what the nodes are — are they mice, features, clusters of mice, clusters of features or none of the above?

Response: Apologies for lack of clarity, we have now modified the caption to clarify that the nodes represent regional mitochondrial features (rows in 1e) that are similar across mice.

A challenge in using the data visualization in 1e,f is that if the reader lacks clear expectations as to what a null model would produce — or is unfamiliar with this particular TDA tool — then they can prove inscrutable. I’d find the conclusions in 1g more convincing if the data were shuffled in an appropriate manner and comparisons to this null made.

Response: We agree with the reviewer and as mentioned above, we have now run appropriate graph null models to quantify the statistical robustness of these models. The p values for these graphs are 0.037 for CORT and 0.069 for CSDS, despite the low number of animals relative to the number of brain regions included.

The authors use Pearson's r — why not use Spearman's rank?

Response: Spearman's rank was used when measuring the correlation between mitochondrial measures and behaviors (Figure 2). In the manuscript this analysis was mislabeled as Pearson's r, although it was correctly labeled as Spearman's r in the figure legend and statistical analysis section. We have corrected this error.

Pearson's r was used to compare mitochondrial measures across all brain regions and for multislice detection (Figure 3). Pearson's r was more appropriate for these analyses because we expected linear relationships between brain regions, instead of monotonic relationships. Previous work on multislice community detection also used similar Pearson's r based-approaches before (Cole et al., 2014; Sarzynska et al., 2015; Sporns and Betzel, 2016).

“Previous research in rodents showed that mitochondrial function in specific brain regions, such as the NAc, is linked to complex behaviors such as social dominance and anxiety^{4,29}. Here we confirm the strong correlation between NAc mitochondrial health and anxious behavior, as the strongest correlation between EPM-based anxiety-like behavior and MHI was in the NAc.”

To me this seems an important corroborative result perhaps this can be emphasised more earlier in the text?

Response: We agree that this is an important result, and thus we now emphasize it also in the results section in addition to the discussion (p.11, p.16).

“the data matrix for each group was process through the TDA-based Mapper pipeline ”

Response: Thank you, we have fixed the typo.

Reviewer #4 (Remarks to the Author):

Mitochondria are essential to proper nervous system functioning with increasing evidence of their functional diversity across distinct cell types, physiologic coupling to neuronal activity, and deficits in neuropsychiatric disease. In this study, Rosenberg, Saggar et al. sought to determine associations between brain region-specific mitochondrial function and animal behavior.

The authors obtained biochemical measurements of mitochondrial respiratory chain function and proxies for mitochondrial content from 17 brain regions of 27 adult male mice that underwent standard behavioral phenotyping. To increase behavioral diversity, 17 of these mice were exposed to models of chronic stress prior to behavioral testing. They identified some significant correlations between mitochondrial functioning in certain brain regions with behavioral metrics (e.g. their derived mitochondrial health index score in nucleus accumbens is positively correlated with anxiety-like behavior across animals ascertained by the elevated plus maze assay). Network analyses on this mitochondrial functional data further showed clusters of brain regions that mirror known neuronal functional networks (e.g. cortico-striatal) and appeared to overlap with mouse structural connectivity and gene coexpression networks.

Overall, the authors have generated a novel dataset providing intriguing associations between mitochondrial function and neurobiological output (i.e., animal behavior). However, it is difficult to interpret the biological meaning of these associations and their ultimate validity given limited follow-up experiments and validation of the original dataset.

Response: We appreciate this reviewer's enthusiasm for the work and the questions it raises.

Suggestions for improvements:

1) A more thorough investigation on the possible biological underpinning of the variance in mitochondrial functioning observed across brain regions would strengthen the study. As the authors mention, there is large cellular compositional differences across the 17 analyzed brain regions with distinct neuronal types and varying proportions of glial cells. As there appears to be brain cell-type specific mitochondrial functional differences (e.g. Fecher et al. Nat Neuro 2019), could part of the differences observed here be based on variation in neuronal/glial composition? To investigate the contribution of cellular composition, one could use existing single-cell RNA-seq atlases from mouse to deconvolute the Allen Brain mouse gene expression atlas used here to get estimates of the major cell type abundances in these 17 brain regions and look for associations to mitochondrial functioning. Similarly, they could use the Allen Brain mouse atlas to look for correlations with regional gene expression to nominate putative biological pathways implicated in the mito-networks (e.g. could further investigate the neuroendocrine hypothesis stated in the discussion). The authors also suggest metabolic coupling with neuronal activity as another possible mechanism. Although technically challenging, it would be interesting to directly test this idea using neuronal activation/inhibition strategies.

Response: The design of our assays, particularly the partitioning of samples into batches (1 batch = 96-well plate with a maximum of 32 samples, in triplicates), were optimized to detect differences between treatment groups, and to avoid the potential confound of batches in the analyses of mouse-to-mouse differences in mitochondrial metrics, with behavior. As a result, we put less confidence in the direct comparisons between brain regions in this manuscript, and consequently have avoided reporting these analyses in the manuscript, instead focusing on outcomes for which the study design is optimal.

To address the reviewer’s comment, we have obtained data for the various cell type abundances across each brain region from the Blue Brain Cell Atlas (Ero et al., 2018; Keller et al., 2018), including total cell density, neuronal density, and neuroglial densities. Specifically, we also included densities of excitatory, inhibitory, and modulatory cells in addition to densities of oligodendrocytes, astrocytes, and microglia. We correlated the density measures for each cell type from each brain region with the average (pooled average across all animals) of each mitochondrial measure for each region. The figure below shows the direction and strength of the correlations. Overall, we find that the proportion of certain cell types are correlated with specific mitochondrial measures (heatmap region highlighted in yellow). For example, the strongest correlation shows that brain regions with a greater proportion of astrocytes tend to exhibit higher citrate synthase (CS) activity ($r^2=0.58$, $p<0.001$), a marker of mitochondrial content (correlation A, *right*). Regions with more oligodendrocytes (Oligos) tend to exhibit greater respiratory chain complex II activity ($r^2=0.48$, $p=0.003$; correlation B, *right*). However, there were no association between the abundance of any cell type and MHI, our composite marker of RC capacity (last column on the right of the heatmap).

The sample size for these comparisons ($n=16$ regions) is fairly limited, particularly in light of the multiple correlations. These correlational data also do not provide direct evidence on the mitochondrial phenotypes of different cell types. It may, however, contribute useful information to this paper. Should the reviewer find these data relevant and deemed them worthy of inclusion in the manuscript, we would be happy to fulfill this request.

2) Experimental validation of select findings would strengthen the evidence of the data collected in the initial dataset. For example, orthogonal experiments to confirm associations of amygdala/PAG mitochondrial function and chronic stress models; NAc mitochondria function and anxiety-like behavior; SN mitochondria function and social interaction. The presented results would also be more convincing if an identified association was shown to be causally linked (as done in prior work, Hollis PNAS 2015).

Response: Yes, prior work by co-author Sandi and others (Gebara et al., 2021; Hollis et al., 2015; van der Kooij et al., 2018) strongly implicate mitochondrial RC activity in the NAc as a driver of social behavior and anxiety. The main contribution of the current work is to perform multiple direct biochemical measures of mitochondrial OxPhos function across a large selection of theoretically-motivated brain regions, and systematically compare their association with animal behavior. This pattern-discovery approach builds on the notion that complex behaviors (and cognition) are driven not by isolated brain regions but by networks of interacting regions (Cole et al., 2014; Ju and Bassett, 2020; Mann et al., 2021; Pope et al., 2021; Sporns and Betzel, 2016). Based on this theoretical foundation, performing validation biochemical experiments as in Hollis et al. by targeting a single brain region would be inconsistent with the overall message and future directions that this work calls for. In addition, while previous work (Gebara et al., 2021; Hollis et al., 2015; van der Kooij et al., 2018) directly reflected mitochondrial features in nucleus accumbens with specific behavioral components, the current study goes beyond the focus on specific brain regions and on specific behaviors to consider more generally how mitochondrial features across multiple brain regions relate to behavioral predispositions across multiple tasks.

Thus, although we agree that testing causal claims linking brain networks to behavior would be particularly convincing, the approach is not currently straightforward or feasible as it would simultaneously target multiple brain regions, and/or involve high temporal and spatial-resolution imaging of functional mitochondrial phenotypes in awake and behaving animals. We hope that the reviewer will agree that the conceptual novelty and added scientific value is sufficient to warrant publication of these results.

3) If I understand the presentation of the results correctly, the vast majority of the associations depicted in the heatmaps in Fig. 1a (side note: are these values corrected for multiple testing?) and Fig. 2c are non-significant. If this is the case, I find the descriptions of the data in the results section a bit odd. E.g. "CORT-treated mice had higher mitochondrial activities than non-stressed animals in ~60% of brain regions, whereas CSDS animals had lower activities in ~82% of brain regions"; but based on Fig.1a only 1 region with 1 metric (out of possible 102 comparisons) was significantly increased in CORT-treated mice. Another example: "For both OFT and EPM, higher mitochondrial health in the majority of brain regions was correlated with higher avoidance scores (average $r=0.12, 0.34$ respectively)"; but only 3 associations in Fig 2c were significantly correlated. In general, how are we supposed to interpret non-significant associations?

Response: The main correlations of mitochondrial measure with behaviors in Figure 2c have been corrected for multiple testing with an adjusted p value of <0.002 (false-discovery rate 1%). Please, note that our study was designed to detect global correlation patterns rather than having adequate power to test every possible combination of brain regions and behavior. Similarly, most human brain neuroimaging studies with hundreds of individuals are well-powered to detect distributed multivariate activity patterns (using multivariate or graph theory metrics, as we do in Figure 3 on data from Figure 2), but are not powered to perform brain-wide univariate association studies (which require thousands of individuals) (Marek et al., 2022).

Using the following parameters for our study: two-tailed $\alpha = 0.05$ (type I error rate) and a $\beta = 0.20$ (type II error rate), the study would require a sample size of 543 animals for a correlation of 0.12 to be significant. With our current study design including 17 brain regions and 5 primary mitochondrial measures, it would not be technically possible or ethical to achieve this sample size. We have toned

down the description of the results to highlight the trends and avoid statements that imply the existence of statistically significant results. The revised text on p.9 now reads as follows: “Although not statistically significant, CORT-treated mice tended to have higher mitochondrial activities than non-stressed animals in ~60% of brain regions, whereas CSDS animals trended towards lower activities in ~82% of brain regions, which was statistically significant for CI, CIV, and MHI measures (Figure 1d).” As indicated in the figure legend, Figure 1a lists % differences only for correlations that have an uncorrected p value <0.05. We hope that these changes avoid confusion and clearly portray the data.

4) Related to point (3), perhaps a clearer presentation of the statistical methods used would help clarify the findings. E.g. highlighting statistically significant associations (and how that’s defined) for each of the heatmaps.

Response: As explained in the previous question 3), we now clarify which correlations are significant.

5) Although not my area of expertise, it was not clear to me why mtDNA density would be a ‘more appropriate and generalizable estimate of mitochondrial genome abundance across mouse brain regions’ compared to mtDNAcn. Wondering about how choice of this metric might introduce more/less confounding with cellular compositional differences across brain regions (naively, I would assume mtDNAcn is better corrected for potential compositional differences).

Response: The cellularity (i.e., density of cell bodies per cubic micron) varies widely between brain regions (Ero et al., 2018; Keller et al., 2018), an observation confirmed in our data across the 17 brain regions. Because it is not confounded by the cellularity of the tissue, the parameter “mtDNA density” is likely a more generalizable estimate of mitochondrial genome abundance. In brain tissues with many

dendritic processes but no cell body, the mtDNA_{cn} (i.e., ratio of mtDNA / nDNA) value is positively skewed by the lack of nuclear DNA (very small nDNA denominator, but large mtDNA numerator). Therefore, when comparing brain regions that differ in cellularity, mtDNA density is likely a more appropriate measure. We have clarified this point in the manuscript (p.6, p.18):

“Compared to mtDNA_{cn}, which is confounded by the presence or absence of somata/nuclear genome, mtDNA density was more consistently associated with RC enzymatic mitochondrial phenotypes across all brain regions, and therefore likely represents a more generalizable estimate of mitochondrial genome abundance across mouse brain regions.”

“Moreover, mtDNA_{cn} has previously been assessed across multiple brain regions (Fuke et al., 2011), but our results go beyond these observations in showing that quantifying mtDNA content on a per-cell basis (mtDNA:nDNA ratio) is heavily skewed by cellularity variations across brain regions and not directly related to **RC energy production** capacity. As such, the mtDNA:nDNA ratio (mtDNA_{cn}) is driven by how many cell bodies are present in the tissue, and correlates poorly with either mitochondrial content or RC activity. Therefore, our data reinforce the notion that mtDNA_{cn} on its own is not a valid measure of mitochondrial **phenotypes** (Cayci et al., 2012; Longchamps et al., 2020). In the mouse brain, our findings **suggest** that mtDNA density per unit of volume (rather than per cell) is a more biologically meaningful mitochondrial feature **when comparing brain regions that differ in cellularity.**”

6) It was also unclear why CSDS model would create such a coherent and opposite effect on mitochondrial functioning compared to CORT-treated mice. Furthermore, from the available data presented in Ext. Data Fig. 8 it appears these animals performed similarly on behavioral tests. Not sure how to reconcile the potential disparate effects on mitochondria functioning seen in these models yet similar behavioral outcomes.

Response: Several factors differ between the CORT and CSDS stressors, given their different - pharmacological vs behavioral- nature, length of treatments, and most probably intensity of the manipulations, making it not particularly relevant to draw strong conclusions on the differential effects between stressors. Importantly, variation in all these factors is inherent to our experimental goal and design and targeted to increase potential sources of variation. Note that, when compared against the null hypothesis that different stressors would trigger equivalent brain recalibrations, our results highlight the notion that mitochondrial recalibrations are region-specific and dependent on the stressor modality. We now discuss this point in more details in the discussion (p.16):

“In contrast to CORT, the recalibrations of brain mitochondria to CSDS was more uniform, with the majority of brain regions exhibiting a coordinated reduction in most mitochondrial features. **This difference in mitochondrial recalibrations between both stress models may be driven by several factors. This includes the stressor duration, although the temporal dynamics over which stress-induced mitochondrial recalibrations take place remain poorly defined and will require further focused attention. Differences in the effects of CORT vs CSDS on mitochondria also could be related to their neuroendocrine underpinnings (single hormone for CORT vs multiple physiological neuroendocrine recalibrations for CSDS), regional differences in glucocorticoid and mineralocorticoid receptor density, differences in the neurocircuits activated/suppressed by each stressor, or other factors generally relevant to interpreting chronic stress rodent models.**”

While it is true that these animals performed similarly on behavioral tests, both groups only overlapped in one behavioral test (OFT), where there were no significant differences even between the stressed groups and the naïve group.

References

- Bindra, S., McGill, M.A., Triplett, M.K., Tyagi, A., Thaker, P.H., Dahmouh, L., Goodheart, M.J., Ogden, R.T., Owusu-Ansah, E., K, R.K., et al. (2021). Mitochondria in epithelial ovarian carcinoma exhibit abnormal phenotypes and blunted associations with biobehavioral factors. *Scientific reports* 11, 11595.
- Bonvento, G., and Bolanos, J.P. (2021). Astrocyte-neuron metabolic cooperation shapes brain activity. *Cell Metab* 33, 1546-1564.
- Cayci, T., Kurt, Y.G., Akgul, E.O., and Kurt, B. (2012). Does mtDNA copy number mean mitochondrial abundance? *J Assist Reprod Genet* 29, 855.
- Cole, M.W., Bassett, D.S., Power, J.D., Braver, T.S., and Petersen, S.E. (2014). Intrinsic and task-evoked network architectures of the human brain. *Neuron* 83, 238-251.
- Ero, C., Gewaltig, M.O., Keller, D., and Markram, H. (2018). A Cell Atlas for the Mouse Brain. *Front Neuroinform* 12, 84.
- Fecher, C., Trovo, L., Muller, S.A., Snaidero, N., Wettmarshausen, J., Heink, S., Ortiz, O., Wagner, I., Kuhn, R., Hartmann, J., et al. (2019). Cell-type-specific profiling of brain mitochondria reveals functional and molecular diversity. *Nat Neurosci* 22, 1731-1742.
- Fuke, S., Kubota-Sakashita, M., Kasahara, T., Shigeyoshi, Y., and Kato, T. (2011). Regional variation in mitochondrial DNA copy number in mouse brain. *Biochim Biophys Acta* 1807, 270-274.
- Gebara, E., Zanoletti, O., Ghosal, S., Grosse, J., Schneider, B.L., Knott, G., Astori, S., and Sandi, C. (2021). Mitofusin-2 in the Nucleus Accumbens Regulates Anxiety and Depression-like Behaviors Through Mitochondrial and Neuronal Actions. *Biol Psychiatry* 89, 1033-1044.
- Gourley, S.L., and Taylor, J.R. (2009). Recapitulation and reversal of a persistent depression-like syndrome in rodents. *Curr Protoc Neurosci* Chapter 9, Unit 9 32.
- Guaras, A., Perales-Clemente, E., Calvo, E., Acin-Perez, R., Loureiro-Lopez, M., Pujol, C., Martinez-Carrascoso, I., Nunez, E., Garcia-Marques, F., Rodriguez-Hernandez, M.A., et al. (2016). The CoQH2/CoQ Ratio Serves as a Sensor of Respiratory Chain Efficiency. *Cell Rep* 15, 197-209.
- Hollis, F., van der Kooij, M.A., Zanoletti, O., Lozano, L., Canto, C., and Sandi, C. (2015). Mitochondrial function in the brain links anxiety with social subordination. *Proc Natl Acad Sci U S A* 112, 15486-15491.
- Ju, H., and Bassett, D.S. (2020). Dynamic representations in networked neural systems. *Nat Neurosci* 23, 908-917.

Keller, D., Ero, C., and Markram, H. (2018). Cell Densities in the Mouse Brain: A Systematic Review. *Front Neuroanat* 12, 83.

Lechuga-Vieco, A.V., Latorre-Pellicer, A., Johnston, I.G., Prota, G., Gileadi, U., Justo-Mendez, R., Acin-Perez, R., Martinez-de-Mena, R., Fernandez-Toro, J.M., Jimenez-Blasco, D., et al. (2020). Cell identity and nucleo-mitochondrial genetic context modulate OXPHOS performance and determine somatic heteroplasmy dynamics. *Sci Adv* 6, eaba5345.

Longchamps, R.J., Castellani, C.A., Yang, S.Y., Newcomb, C.E., Sumpter, J.A., Lane, J., Grove, M.L., Guallar, E., Pankratz, N., Taylor, K.D., et al. (2020). Evaluation of mitochondrial DNA copy number estimation techniques. *PLoS One* 15, e0228166.

Lopez-Fabuel, I., Le Douce, J., Logan, A., James, A.M., Bonvento, G., Murphy, M.P., Almeida, A., and Bolanos, J.P. (2016). Complex I assembly into supercomplexes determines differential mitochondrial ROS production in neurons and astrocytes. *Proc Natl Acad Sci U S A* 113, 13063-13068.

Mann, K., Deny, S., Ganguli, S., and Clandinin, T.R. (2021). Coupling of activity, metabolism and behaviour across the *Drosophila* brain. *Nature* 593, 244-248.

Marek, S., Tervo-Clemmens, B., Calabro, F.J., Montez, D.F., Kay, B.P., Hatoum, A.S., Donohue, M.R., Foran, W., Miller, R.L., Hendrickson, T.J., et al. (2022). Reproducible brain-wide association studies require thousands of individuals. *Nature* 603, 654-660.

Picard, M., Prather, A.A., Puterman, E., Cuillerier, A., Coccia, M., Aschbacher, K., Burelle, Y., and Epel, E.S. (2018). A mitochondrial health index sensitive to mood and caregiving stress. *Biol Psychiatry* 84, 9-17.

Pope, M., Fukushima, M., Betzel, R.F., and Sporns, O. (2021). Modular origins of high-amplitude co-fluctuations in fine-scale functional connectivity dynamics. *Proc Natl Acad Sci U S A* 118.

Rausser, S., Trumpff, C., McGill, M.A., Junker, A., Wang, W., Ho, S.H., Mitchell, A., Karan, K.R., Monk, C.E., Segerstrom, S.C., et al. (2021). Mitochondrial phenotypes in purified human immune cell subtypes and cell mixtures. *eLife* 10.

Sarzynska, M., Leicht, E.A., Chowell, G., and Porter, M.A. (2015). Null models for community detection in spatially embedded, temporal networks. *Journal of Complex Networks* 4, 363-406.

Sporns, O., and Betzel, R.F. (2016). Modular Brain Networks. *Annu Rev Psychol* 67, 613-640.

Stepanova, A., Shurubor, Y., Valsecchi, F., Manfredi, G., and Galkin, A. (2016). Differential susceptibility of mitochondrial complex II to inhibition by oxaloacetate in brain and heart. *Biochim Biophys Acta* 1857, 1561-1568.

van der Kooij, M.A., Hollis, F., Lozano, L., Zalachoras, I., Abad, S., Zanoletti, O., Grosse, J., Guillot de Suduiraut, I., Canto, C., and Sandi, C. (2018). Diazepam actions in the VTA enhance social dominance and mitochondrial function in the nucleus accumbens by activation of dopamine D1 receptors. *Mol Psychiatry* 23, 569-578.

Xie, X., Shen, Q., Yu, C., Xiao, Q., Zhou, J., Xiong, Z., Li, Z., and Fu, Z. (2020). Depression-like behaviors are accompanied by disrupted mitochondrial energy metabolism in chronic corticosterone-induced mice. *J Steroid Biochem Mol Biol* 200, 105607.

REVIEWER COMMENTS

Reviewer #1 (Remarks to the Author):

The authors have successfully addressed all the concerns raised by this reviewer and therefore there are no further comments as the manuscript is much improved with the clarifications and discussions added. Congrats on this very nice piece of work.

Reviewer #2 (Remarks to the Author):

In their response to reviewer comments the authors have adequately addressed all reviewer concerns raised.

Reviewer #3 (Remarks to the Author):

Thank you for responding to my comments. I think it is important to tell the reader, for each case, whether the results have been corrected for testing for multiple hypotheses or not. My strong preference would be that this should not be buried in a supplement or methods section but be clear in e.g. the appropriate figure caption or associated piece of main-text.

Reviewer #4 (Remarks to the Author):

The revised manuscript by Rosenberg and colleagues contains mostly textual clarifications, a few added analyses (e.g. a null network for Fig. 1), and 1 new extended figure with new data validating the enzymatic assay. Overall, I feel this is a rather minor revision attempt to address the various suggestions put forth by the reviewers. While there appears to be enthusiasm for the brain region based mitochondrial feature dataset, as the authors acknowledge, many more samples would be needed to make robust statistical claims on associations between specific brain regions and behavior/treatment (i.e. Fig 1a, Fig 2c). In general, I would favor de-emphasizing all results that are not adequately powered and focusing on the more robust aggregated data (i.e. Fig 1c,d; Fig. 2b,d; as suggested by reviewer 3) and network analyses of Figure 3, which are more compelling and interpretable.

Further suggestions for improvement:

I still would suggest a more thorough accounting for the statistical comparisons. Perhaps a supplementary table with all tested differences might be useful. Why are unadjusted p-values highlighted and presented in some places (e.g. in the text, Fig 1a, Fig 1b)? Why are non-significant 'trend' p-values reported (e.g. Fig 1g)?

I do not find Figure 1a, Fig e-f, or Fig 2c compelling (i.e. multiple non-significant tests, low effects, and possibly underpowered to make conclusions).

Perhaps an alternative way of presenting Fig 1f could help to visualize the conclusion: "CORT-induced mitochondrial recalibrations were relatively more region-specific or segregated, whereas CSDS caused a more coherent or integrated mitochondrial response across all brain regions".

The brain-wide mitochondrial connectivity results (Fig 3) and potential associations with behavior are interesting. However, correlation values in result section and Figure 3g are presented without statistical tests.

Appreciated the attempt to account for contributions of mitochondrial feature variation due to neuron/glia compositional differences by brain region. I think this would add to the paper if included.

I still think the paper would be strengthened by more deeper analyses looking at the overlap with

gene coexpression and structural connectome data. For example, are the “similar communities” that overlap across comparisons with gene co-expression networks or structural connectivity driven by certain modules of genes or specific regional couplings? This would be more of an exploratory analysis, but could get at possible biological interpretations of the identified mito networks, and suggest future follow-up studies.

Reviewer #1 (Remarks to the Author):

The authors have successfully addressed all the concerns raised by this reviewer and therefore there are no further comments as the manuscript is much improved with the clarifications and discussions added. Congrats on this very nice piece of work.

R: Thank you

Reviewer #2 (Remarks to the Author):

In their response to reviewer comments the authors have adequately addressed all reviewer concerns raised.

R: Thank you

Reviewer #3 (Remarks to the Author):

Thank you for responding to my comments. I think it is important to tell the reader, for each case, whether the results have been corrected for testing for multiple hypotheses or not. My strong preference would be that this should not be buried in a supplement or methods section but be clear in e.g. the appropriate figure caption or associated piece of main-text.

R: As requested, this information is now noted in each figure caption.

Reviewer #4 (Remarks to the Author):

Comment 1: The revised manuscript by Rosenberg and colleagues contains mostly textual clarifications, a few added analyses (e.g. a null network for Fig. 1), and 1 new extended figure with new data validating the enzymatic assay. Overall, I feel this is a rather minor revision attempt to address the various suggestions put forth by the reviewers. While there appears to be enthusiasm for the brain region based mitochondrial feature dataset, as the authors acknowledge, many more samples would be needed to make robust statistical claims on associations between specific brain regions and behavior/treatment (i.e. Fig 1a, Fig 2c). In general, I would favor de-emphasizing all results that are not adequately powered and focusing on the more robust aggregated data (i.e. Fig 1c,d; Fig. 2b,d; as suggested by reviewer 3) and network analyses of Figure 3, which are more compelling and interpretable.

R1: We appreciate the reviewer's thoughtful comments. Given that this is the first report of multiple mitochondrial enzymes across a large number of brain regions, we view this novel data on individual brain areas as foundational. These results answer more basic (and we agree, possibly less robust) questions that we hope to serve as a foundation for future research bridging mitochondrial-level analyses and behavioral outcomes. For example, if one knows by what percentage a mitochondrial enzyme changes in response to chronic CORT or social defeat stress in a given brain region (Fig 1a), plus the effect sizes for activity-behavior associations (Fig 2c), and also the inter-individual variation (Extended Data Fig 5: C.V. reported by brain region, for each enzyme), then one can run power calculations and design a study to address specific mito-behavior hypotheses.

The reality is, as illustrated by the heatmaps, that mitochondria in different brain areas respond to stress and are associated with behavior in remarkably different ways. Moreover, different mitochondrial features (specific respiratory chain enzymes or content vs. respiratory chain features) respond in ways that may not be consistent with other features (owing to biological differences in their regulation, molecular assembly, or other factors). We fear that removing or de-emphasizing these results may

convey an overly simplistic picture, which albeit more palatable or digestible for most readers, may not be the most scientifically useful.

In response to the reviewer's comments below, this revision includes additional analyses from the Allen Mouse Brain Atlas (Figure 4) that define the molecular characteristics of the identified mito-based brain networks and supports our network findings, which we hope adds further value to this manuscript.

Further suggestions for improvement:

Comment 2: I still would suggest a more thorough accounting for the statistical comparisons. Perhaps a supplementary table with all tested differences might be useful. Why are unadjusted p-values highlighted and presented in some places (e.g. in the text, Fig 1a, Fig 1b)? Why are non-significant 'trend' p-values reported (e.g. Fig 1g)?

R2: The revised manuscript thoroughly accounts for all statistical comparisons, stating in each figure legend and in the main text where appropriate, where unadjusted p values are used. As mentioned above, some fine-grained analyses are necessarily underpowered, so the standardized effect sizes (Hedges' g) provide a more useful estimate of the true effect. We also provide the p values more systematically to make the results as transparent as possible.

Comment 3: I do not find Figure 1a, Fig e-f, or Fig 2c compelling (i.e. multiple non-significant tests, low effects, and possibly underpowered to make conclusions).

R3: Figure 1a will most likely be useful for investigators interested in stress-induced mitochondrial recalibrations. As noted above, this panel illustrates the heterogeneity in the observed recalibrations across brain regions and non-brain tissues in the most transparent possible way. Figure 1e seems necessary to help readers appreciate the underlying analytical structure, and the meaning of Figure 1f is now clarified by the statement helpfully suggested below. Without showing Figure 2c, most readers may not readily grasp the nature of the data underlying subsequent analyses, so we have respectfully preserved these panels in the figures.

Comment 4: Perhaps an alternative way of presenting Fig 1f could help to visualize the conclusion: "CORT-induced mitochondrial recalibrations were relatively more region-specific or segregated, whereas CSDS caused a more coherent or integrated mitochondrial response across all brain regions".

R4: Thank you. This statement has been emphasized in the text, and we have more clearly outlined the statistical significance in Fig 1f by adding the p-value directly on the graph.

Comment 5: The brain-wide mitochondrial connectivity results (Fig 3) and potential associations with behavior are interesting. However, correlation values in result section and Figure 3g are presented without statistical tests.

R5: We re-computed these correlations by first aggregating mitochondrial features at the regional level, followed by aggregating values across all regions in each network, then computing correlations with behaviors. These results are now clearly presented in Figure 3g (* p<0.05, **p<0.01) and noted in the figure legend. The new calculations demonstrate the same effects, but the effect sizes are slightly larger than our previous estimates, for which we had earlier reported average mito-behavior correlation coefficients across brain areas within each network. To compute statistics, now we first pool data across regions and then run correlations with behavior at the network level. The correlations for 3 of the 4 behavioral tests show significant correlations (p<0.05 or p<0.01) with network 1 mitochondrial features,

shown both at the network aggregate-level in the table (top of 3g), in graph form with each node and network highlighted (middle), and with the individual scatterplots (bottom).

Additionally, for figure 3A, we have performed FDR correction on p-values to identify the significant correlations. We performed False-Discovery Rate (FDR) correction using the linear step-up procedure introduced by Benjamini and Hochberg (1995) to obtain FDR-corrected p-values. The figure below shows the FDR-corrected Pearson correlation coefficients at $p < 0.05$. This result highlights three main points. First, as discussed above, we are underpowered to detect statistical significance for the number of comparisons performed here (matrix of 22 x 22 tissues). Second, as discussed in the previous version of the manuscript, the correlations between brain regions are generally weak. In this underpowered scenario, they do not survive FDR correction. This is consistent with our previous interpretation. Third, although the patterns of correlations at the individual brain area level discernable in the main Figure 3a are informative (and the first of its kind for mitochondrial measures in the brain), it is a more robust approach here to focus on the distribution patterns of correlations. Thus, this reinforces our downstream strategy to use multi-slice community detection to identify networks of brain areas related to multiple mitochondrial metrics.

Comment 6: Appreciated the attempt to account for contributions of mitochondrial feature variation due to neuron/glia compositional differences by brain region. I think this would add to the paper if included.

R6: This analysis showing the correlations of mitochondrial features (per mg of brain tissue) and cell counts (cells per μm^2) is now included as Extended Data Figure 11a-c (below). While this information is valuable, we note that the results are subject to the caveat that some brain areas have outlier values for the number of neurons relative to other brain areas, such as the cerebellum that contains approximately 10 times more cell nuclei per tissue area than other areas. This makes the regression coefficients across the 17 areas less reliable.

Therefore, in panels d-f, we compute the proportion (%) of cells that are either neurons or glia, as well as % of neuronal and glial subtypes. This takes care of the outlier values and reflects more specifically the cell type distribution across brain areas. We also normalize all mitochondrial features by the number of cell nuclei (i.e., total cellularity) for each area. This combines the effects of both mitochondrial content/energy transformation capacity, relative to the number of cell bodies. This new analysis shows that areas with many *neuronal* cell bodies tend to have few mitochondria and lower mtDNA density per cell nucleus. On the other hand, areas with many glial cell bodies have higher mitochondrial enzyme activities on a per cell basis. This finding may be consistent with the fact that the bulk of mitochondrial mass is located in the dendrites and distant arbors, supported by glial cells, rather than in neuronal cell bodies.

A new statement in the text on p.13 describes these results:

“To estimate how much the observed variance of mitochondrial features across brain areas may be driven by variations in cell type composition, we correlated the abundance and proportion of various cell types including neuronal and glial cell subtypes (available from the Blue Brain Cell Atlas^{61,62}), with the average value of each mitochondrial feature (across all animals in our cohort). Of the brain areas sampled in this study, those with more astrocytes and oligodendrocytes had higher mitochondrial enzyme activities (**Extended Data Figure 11**). However, on average MHI was not correlated with any of the cell subtypes, indicating that in the mouse brain the variation in mitochondrial phenotypes between areas is only partially driven by differences in cellular composition.”

The new Extended Data Figure 11:

Legend: Extended Data Fig. 11. Correlation of cell type densities and mitochondrial features across brain areas. (a) Data on the abundance or density of various cell types was extracted for each area (n=16 areas, dorsal and ventral not differentiated) from the Blue Brain Cell Atlas^{61,62}, and correlated with the average (pooled average across all animals) of each mitochondrial feature. The green box indicates the associations between cell type abundances and mitochondrial features. For each brain area, MHI is computed from the average mitochondrial features across all animals. (b) Scatterplots for the strongest observed correlations between astrocyte density and citrate synthase activity, and (c) oligodendrocyte density (Oligos) and complex II activity. (d) Correlation between the proportion (%) of cell types/subtypes and mitochondrial features normalized to total density of cells for each brain area (i.e., cell count or cellularity). This normalization combines the influence of both mitochondrial content/energy transformation capacity, relative to the number of cell bodies, such that areas with few cell bodies but many mitochondria-rich dendritic arbors have the highest normalized mtDNA density and enzyme activities. (e, f) Scatterplots for the strongest correlations from panel d. Abbreviations: AP axis, anterior-posterior axis; mtDNA, mtDNA density (relative copies per unit of brain mass).

Comment 7: I still think the paper would be strengthened by more deeper analyses looking at the overlap with gene coexpression and structural connectome data. For example, are the “similar

communities” that overlap across comparisons with gene co-expression networks or structural connectivity driven by certain modules of genes or specific regional couplings? This would be more of an exploratory analysis, but could get at possible biological interpretations of the identified mito networks, and suggest future follow-up studies.

R7: To enhance the biological interpretation of the enzyme activity mito-based networks, we performed two sets of analyses presented in the new **Figure 4**.

First, we performed an unbiased analysis of gene expression differences between the three identified networks. This analysis establishes whether there are shared molecular features among all areas of a given network that differentiates it from other networks. To do this, we used the Allen Mouse Brain Atlas, surveyed all microscopic areas with *in situ* gene expression data (n=2,170), and averaged the microscopic areas that correspond to each of our target brain areas (details in a new **Supplemental File 1**). The Allen Brain Atlas did not contain data differentiating dorsal vs ventral dentate gyrus, so this analysis proceeded with n=16 areas (instead of 17 with our tissue punches). We then collapsed all areas of each of the network 1 (n=6 areas), network 2 (6 areas), and network 3 (4 areas) to create average gene expression values for each network. To compare networks to one another, we used a threshold of one Log2 unit, yielding genes either *overexpressed* by >100% (double) or *underexpressed* by >50% (half) in one network relative to the other two networks. Given that all samples are areas of the same brain tissue, this is a relatively stringent cutoff. Using the resulting list of genes either over- or under-expressed in each network (provided in a new **Supplemental File 2**), we performed gene ontology (GO) analysis to ask which biological processes are enriched (FDR p<0.05) in each mito-based brain network, and then used a simple network-based approach to group the resulting biological processes into a few simple and interpretable categories of differentially expressed biological pathways. Because our main findings linking mitochondrial activities and behavior in Figure 3 relate to network 1 (strongest correlation between mitochondrial activities and behavior), we report the enriched categories for Network 1 relative to the other two networks in the main figure. The results for the networks 2 and 3 in the new **Extended Data Figure 13**. The results show that the cortico-striatal network 1 brain areas overexpress genes related to synaptic signaling and transmission, neuronal and dendritic morphogenesis, and enzyme regulation by phosphorylation. This is a markedly different picture than network 2 and 3. This analysis adds further specificity and molecular information about the nature of the molecular features that differentiate network 1 areas from other brain areas examined in our study.

In the second analysis, we refine this approach by focusing on differences in mitochondrial phenotypes specifically. To do this, we focus our analysis on mitochondrial genes only (n=946, MitoCarta3.0, Rath et al. *Nucl Acid Res* 2021). This allows us to ask which kind of mitochondrial transcriptional programs are active, which in turn determine the mitochondrial phenotype (i.e., mitotype) of each brain area. The null hypothesis is that mitochondria across different brain areas roughly have the same molecular composition, and that only their total amount varies (i.e., more or less of all mitochondrial transcripts). Contrary to this hypothesis, we find that the mitochondrial phenotypes of our 16 brain areas vary considerably, and cluster in a non-random way. All areas that compose network 1 cluster together – either in a 3D-PCA space, or by hierarchical clustering based on 149 mitochondrial pathways – meaning that they share a similar transcriptional mitochondrial phenotype. This is indicative of a similar functional specialization. This may not seem surprising given that the networks were generated based on mitochondrial enzymatic activities and mtDNA. However, there are several molecular regulatory steps between gene expression data and enzyme activities. More importantly, these *in situ* hybridization RNA data are from the Allen Mouse Brain Atlas (different mice, ages, etc), and thus completely independent from our specific mouse cohort. Thus, these independent data add further confidence in the robustness of the identified mito-based brain networks. Using a new approach to functionally interpret mitotypes, we also learn from this analysis that relative to networks 2/3, the mitochondria in network 1 brain areas transcriptionally specialize for certain biological pathways, and show a relative

downregulation of other mitochondrial pathways consistent with their anatomical locations. The nature of these differences is described in detail in a new section of the results related to **Figure 4**, and reproduced below for convenience. Thus, these new analyses add further depth to the mapping of mitochondrial phenotypes across the mouse brain and reinforce our major conclusion: distinct mitochondria phenotypes are regionally distributed according to behaviorally-relevant large-scale networks across the mouse brain.

We have added corresponding new sections of text in the methods and results to describe these new analyses. The revised version of this manuscript also includes an interactive R markdown file (**Supplemental File 3**) where readers can explore these results in more depth, for example by hovering over interactive graphs, manipulate the 3D PCA plots of mitochondrial gene expression signatures, and consult the source code used in all analyses. This is the most transparent way to report these findings and to empower others to consult and expand upon these exciting findings.

New Figure 4:

New results section (p.16-18):

Validation of brain networks by transcriptional mitochondrial phenotypes

Having established that the examined 17 brain areas empirically cluster as networks based on their mitochondrial enzyme activities and mtDNA features, we then sought to i) examine the molecular specificity of the most behaviorally-relevant cortico-striatal network 1, and ii) to test whether a different data modality (gene expression), among a different animal cohort of the same strain, could provide converging evidence that the network 1 mitochondrial phenotype qualitatively and quantitatively differ from that of other brain areas.

Therefore, we integrated the *Allen Mouse Brain Atlas* gene expression data for each area, averaged by network (see *Materials and Methods*; n=16 areas, dorsal and ventral DG are combined in the reference dataset). We then identified genes that were on average over- or under-expressed by at least a factor of 2 (double, or half) in network 1 areas relative to all other areas (Network 2+3). Relative to other brain areas, network 1 was significantly enriched for processes related to synaptic signaling and transmission, neuronal and synaptic morphogenesis, and enzyme regulation by phosphorylation (**Fig. 4a**). In contrast, under-expressed network 1 genes were involved in metabolic processes, oxygen and chemical sensing, and anion membrane transport were. Networks 2 and 3 showed remarkably orthogonal gene expression signatures highlighting upregulation of intracellular calcium regulation, extracellular matrix organization, and response to hormonal signaling, among others genetic pathways (**Extended Data Fig. 13**).

Based on evidence that mitochondria functionally and molecularly specialize between tissues and cell types¹⁴⁻¹⁹, we then performed a similar analysis restricted to mitochondrial genes only¹⁶. The resulting mitochondrial gene expression signatures for each brain area projected on

a 3D principal component analysis (PCA) revealed remarkably diverse molecular mitochondrial phenotypes – or *mitotypes*³², where all cortico-striatal network 1 areas clustered together without overlap with network 2/3 areas (**Fig. 4b**). This means that different brain areas express relatively unique *mitochondrial* molecular programs, which are relatively homogenous or shared among all network 1 areas.

We further examined which mitochondrial pathways distinguished each brain area by computing mitochondrial pathway scores, using gene-to-pathway annotations from MitoCarta3.0¹⁶. Pathway-level analysis corroborated our biochemical findings in two ways. First, network 1 areas shared similar mitochondrial phenotypic signatures (i.e., tended to cluster together in unsupervised analysis, right of heatmap). Second, as in our genome-wide analysis, several metabolism-related and energy transformation mitochondrial pathways were under-expressed among network 1 brain areas (**Fig. 4c**). In contrast, we found that the vestibular nucleus, a brainstem area that preferentially degenerates in mouse models of complex I defects (whereas other cortical and subcortical areas appear spared)^{68,69}, expressed the highest level of OxPhos components among all brain areas analyzed (**Fig. 4c**, left column). This finding possibly provides a heretofore missing molecular basis for the preferential vulnerability of the brainstem to mitochondrial OxPhos defects, but more generally highlighted the diversity and specialization of mitochondrial phenotypes across the mouse brain.

Mitochondrial phenotypes (mitotypes) differ across brain areas

Finally, we asked which biological functions differed, and by how much, between network 1 mitochondria versus other brain areas. We ranked the fold differences of mitochondrial pathway scores between network 1 vs network 2+3, from the lowest to highest (**Fig. 4d**). Effect sizes were small, in part due to the substantial variation among the heterogeneous network 2+3 areas that are averaged for these analyses. Network 1 mitochondria were specialized or enriched for the glycerol-phosphate (G3P) shuttle, which transfers cytoplasmic reducing equivalents (NADH) to the quinone (Q) pool to by-pass complex I and directly feed the mitochondrial respiratory chain⁷⁰, amidoxime metabolism (molybdenum-containing enzymes that reduce *N*-oxygenated structures⁷¹), and vitamin D metabolism. In contrast, network 1 mitochondria expressed the lowest levels of Vitamin B2 and B6 metabolism enzymes, tetrahydrobiopterin synthesis (BH4, a cofactor in the production of serotonin, dopamine, and nitric oxide⁷²), and the cristae-organizing complex MICOS that physically interacts with RC complexes to enhance OxPhos capacity⁷³. The direction and magnitude of differences between brain areas and networks is illustrated in bivariate mitotype plots (**Fig. 4e, f**) and calculated ratios of pathway scores (**Fig. 4g, h**), illustrating the extent to which mitochondria specialize for specific biological functions among the mouse brain.

These mitochondrial pathway-level analyses in the reference Allen Mouse Brain Atlas dataset indicate that network 1 brain areas exhibit notable molecular divergences from network 2+3. This result in part confirms the divergences identified in biochemical mitochondrial phenotypes, which was the basis to define network 1 in the first place. Thus, this independent analysis of brain molecular mitochondrial phenotypes in an independent animal cohort further supports the existence of the behaviorally-relevant mito-based networks in Figure 3.

New discussion section (p.23):

In relation to the specialization of mitochondrial phenotypes across the mouse brain, the transcriptional signatures from the Allen Mouse Brain Atlas confirmed that brain mitochondria are not all created equal. Our results thereby extend previous work highlighting distinct molecular and functional mitochondrial phenotypes in different brain cell types¹⁴⁻¹⁹, and document several new observations of area-specific mitotypes (see Figure 4c). Understanding the origin and functional significance of mitotype variations, as well as the extent to which such mitotype variation applies to other species including in the human brain, are exciting questions for future research.

New methods section (p.40-42):

Genome-wide differential gene expression analyses on brain networks

Differential gene expression analysis between the three identified networks was performed using the *in situ* hybridization RNA transcript abundance data from the Allen Mouse Brain Atlas^{64,106} accessed via The Harmonizome¹⁰⁷ (<https://maayanlab.cloud/Harmonizome/dataset/Allen+Brain+Atlas+Adult+Mouse+Brain+Tissue+Gene+Expression+Profiles>). Starting from all microscopic areas with *in situ* gene expression data (n=2,170), areas that correspond to each of our target brain areas were averaged to obtain a single expression value for each gene, per brain area (details in **Supplemental File 1**). The atlas data did not differentiate between dorsal and ventral dentate gyrus, so for these analyses the two areas were combined into one. We then collapsed all areas of each of the network 1 (n=6 areas), network 2 (6 areas), and network 3 (4 areas) to create average gene expression values for each network. To compare networks to one another, we used a threshold of one Log2 unit, yielding genes either *overexpressed* by >100% (double) or *underexpressed* by >50% (half) in one network relative to the other two networks. Using the resulting list of genes either over- or under-expressed in each network (**Supplemental File 2**), we performed gene ontology (GO) analysis to ask which biological processes are enriched (FDR p<0.05) in each mito-based brain network, and then used a network-based approach to group the resulting biological processes into a few simple categories of pathways in ShinyGo 0.76.3. We report separately the enriched categories for network 1 relative to networks 2+3 combined (Figure 4a), and for networks 2 and 3 relative to other networks (**Extended Data Figure 13**).

Transcriptional mitochondrial phenotypes and specialization

For the analyses of mitochondrial phenotypes (i.e., mitotypes), we mapped 946 out of 1136 mitochondrial genes listed in MitoCarta 3.0¹⁶ to the Allen Mouse Brain Atlas gene expression dataset. **Supplementary File 3** (R Markdown) includes a full list of mitochondrial genes identified and those that could not be mapped. To compare the mitochondrial gene and pathway signatures between 16 main areas (dorsal and ventral DG combined) from networks 1, 2 and 3, we used *Funny-Looking Kid* in R version 4.2.1. We also performed a test of robustness and sensitivity analysis by repeating these analyses using all microscopic sub-areas that were averaged into our main 16 areas, which provided additional evidence for the distinct mitochondrial phenotypes among network 1 subareas compared to other areas (**Supplementary File 3**). For the pathway-specific analysis, each mitochondrial gene was assigned to a mitochondrial pathway (n=149) using MitoCarta3.0 annotations. The resulting data was z-score transformed with a mean of 100 and a standard deviation of 10 (i.e. a transformed z-score) to allow for direct gene expression comparisons between brain areas. From the transformed data, the expression of genes in a given pathway were averaged, yielding 149 mitochondrial pathway scores for each brain area. Hierarchical clustering (Figure 4c) of the resulting matrix (16 brain areas x 149 pathways) was performed using the Euclidean distance calculated from relative pathway scores and the *ward.D2* method.

For the differential analysis of mito-pathways across networks, a mean mitochondrial pathway score was calculated by taking the average scores for all areas belonging to network 1 and networks 2+3, and log2 fold difference values were used to rank pathways from lowest (lower in network 1) to highest (higher in network 1). Bi-variate plots as in reference³² were used to visualize area-specific transcriptional mitochondrial phenotypes and to quantify the relative magnitude of mitochondrial specialization between brain areas, using two top and two bottom pathways, color-coded by network. A detailed description of the methods, including the code used in the analyses and more detailed interactive plots with subareas of the 16 main analyzed areas, is available as an R markdown file (**Supplementary file 3**).

R Markdown file – Supplemental File 3. The markdown file should have the extension “.html” (the journal submission system may have changed it) and can be opened with any web browser.

REVIEWERS' COMMENTS

Reviewer #4 (Remarks to the Author):

I thank the authors for thoroughly responding to my comments and adding several new analyses, including a new Figure 4 which further demonstrates molecular phenotypes and pathways (gene expression) that are associated with the identified biochemical mitochondrial networks. They have adequately addressed my comments and I believe the manuscript is much improved.

I have one very minor textual suggestion, which would be to define or further explain in the results section for Figure 4 and legend (4b), what "mitochondrial genes" are. I think some readers might initially assume the authors are referring to genes encoded by the mitochondrial genome rather than a curated list by MitoCarta3.0, which appears to define genes with "strong support of mitochondrial localization". This can be done during editorial editing, and I do not need to further review changes for this edit.

NCOMMS-21-28560B

Reviewer #4 (Remarks to the Author):

I thank the authors for thoroughly responding to my comments and adding several new analyses, including a new Figure 4 which further demonstrates molecular phenotypes and pathways (gene expression) that are associated with the identified biochemical mitochondrial networks. They have adequately addressed my comments and I believe the manuscript is much improved.

I have one very minor textual suggestion, which would be to define or further explain in the results section for Figure 4 and legend (4b), what “mitochondrial genes” are. I think some readers might initially assume the authors are referring to genes encoded by the mitochondrial genome rather than a curated list by MitoCarta3.0, which appears to define genes with “strong support of mitochondrial localization”. This can be done during editorial editing, and I do not need to further review changes for this edit.

R: Thank you. In response to the reviewer’s comment, we have clarified this term in the figure legend and in the text (p.17):

“Based on evidence that mitochondria functionally and molecularly specialize between tissues and cell types¹⁴⁻¹⁹, we then performed a similar analysis restricted to mitochondrial genes only. For this, we used MitoCarta 3.0, an inventory of genes that encode proteins localized in mitochondria¹⁶.”